# Nesterov Finds GRAAL: Optimal and Adaptive Gradient Method for Convex Optimization

**Ekaterina Borodich**
Higher School of Economics; Yandex Research
`borodich.ed@phystech.edu`

**Dmitry Kovalev**
Yandex Research
`dakovalev1@gmail.com`

## Abstract

In this paper, we focus on the problem of minimizing a continuously differentiable convex objective function, $\min_x f(x)$. Recently, Malitsky (2020); Alacaoglu et al. (2023) developed an adaptive first-order method, GRAAL. This algorithm computes stepsizes by estimating the local curvature of the objective function without any line search procedures or hyperparameter tuning, and attains the standard iteration complexity $\mathcal{O}(L\|x_0 - x^*\|^2/\epsilon)$ of fixed-stepsize gradient descent for $L$-smooth functions. However, a natural question arises: is it possible to accelerate the convergence of GRAAL to match the optimal complexity $\mathcal{O}(\sqrt{L\|x_0 - x^*\|^2/\epsilon})$ of the accelerated gradient descent of Nesterov (1983)? Although some attempts have been made by Li & Lan (2025); Suh & Ma (2025), the ability of existing accelerated algorithms to adapt to the local curvature of the objective function is highly limited. We resolve this issue and develop GRAAL with Nesterov acceleration, which can adapt its stepsize to the local curvature at a geometric, or linear, rate just like non-accelerated GRAAL. We demonstrate the adaptive capabilities of our algorithm by proving that it achieves near-optimal iteration complexities for $L$-smooth functions, as well as under a more general $(L_0, L_1)$-smoothness assumption (Zhang et al., 2019).

## 1 Introduction

First-order, or gradient, optimization methods are highly popular in many practical applications due to their simplicity and scalability. However, the key limitation of these methods is that they require the choice of the stepsize parameter. In this paper, we focus on developing efficient gradient methods that can adjust the stepsize at each iteration in an adaptive manner. Formally speaking, we consider the following optimization problem:

$$\min_{x \in \mathbb{R}^d} f(x), \tag{1}$$

where $\mathbb{R}^d$ is a finite-dimensional Euclidean space, and $f(x) \colon \mathbb{R}^d \to \mathbb{R}$ is a convex continuously differentiable objective function. We assume that problem (1) has a solution $x^* \in \mathbb{R}^d$.

### 1.1 Gradient Methods

The simplest and most fundamental example of first-order methods is gradient descent (GD). This algorithm performs iterations to find an approximate solution to problem (1) according to the following update rule:

$$x_{k+1} = x_k - \eta \nabla f(x_k), \tag{2}$$

where $\eta > 0$ is the stepsize. Despite its simplicity, GD and its variants are widely used in practice, especially for solving large-scale problems that often appear, for instance, in machine learning. It is well-known (Polyak, 1963; Nesterov et al., 2018; Drori & Teboulle, 2014) that in the case where the objective function $f(x)$ is $L$-smooth, i.e., the gradient $\nabla f(x)$ is $L$-Lipschitz, GD with the stepsize $\eta = 1/L$ achieves the following iteration complexity:

$$K = \mathcal{O}\big(L\|x_0 - x^*\|^2/\epsilon\big) \quad \Rightarrow \quad f(x_K) - f(x^*) \le \epsilon. \tag{3}$$

In addition, in his seminal work, Nesterov (1983) proposed a modification of GD that implements acceleration via momentum. This accelerated gradient descent (AGD) achieves substantially improved iteration complexity:

$$K = \mathcal{O}\big(\sqrt{L\|x_0 - x^*\|^2/\epsilon}\big) \quad \Rightarrow \quad f(x_K) - f(x^*) \leq \epsilon. \tag{4}$$

It was shown that AGD is an optimal algorithm. That is, the complexity in eq. (4) cannot be improved by any first-order optimization method due to the lower complexity bounds of Nesterov et al. (2018).

## 1.2 ADAPTIVE METHODS

One of the main issues with the standard GD and AGD is that they require tuning the stepsize $\eta$. In particular, they require knowledge of the gradient Lipschitz constant $L$ to achieve the iteration complexities in eqs. (3) and (4). A typical approach to addressing this issue is to use a time-varying stepsize $\eta_k$ in eq. (2), which is computed at each iteration according to a certain adaptive rule.

**Line search.** The simplest way to compute the stepsize $\eta_k$ is to use line search (or backtracking), an iterative procedure that finds $\eta_k$ satisfying a certain objective function value decrement condition. It was originally proposed by Goldstein (1962); Armijo (1966), and its modern variant for GD and AGD was analyzed by Nesterov (2013). Unfortunately, line search makes the iterations of gradient methods more expensive as it requires the computation of the gradient $\nabla f(x)$ and/or function value $f(x)$ multiple times without making any "progress". Hence, it is rarely used in practice.

**AdaGrad-type methods.** An alternative approach is to use the following stepsizes $\eta_k$ in eq. (2):

$$\eta_k = \eta \cdot \big(\textstyle\sum_{i=0}^k \|\nabla f(x_i)\|^2\big)^{-1/2}, \tag{5}$$

where $\eta > 0$ is a positive parameter. The resulting algorithm is called AdaGrad and was originally developed by Duchi et al. (2011); McMahan & Streeter (2010). It is well known that AdaGrad has the complexity of GD in eq. (3) for $L$-smooth functions (Levy et al., 2018). Moreover, this algorithm is universal: it can also achieve the corresponding complexities of GD for non-smooth but Lipschitz functions, or in the case where only stochastic estimates of the gradients are available, all with a single choice of the parameter $\eta \propto \|x_0 - x^*\|$ (Levy et al., 2018; Li & Orabona, 2019; Orabona, 2023). In addition, accelerated variants of AdaGrad are available (Levy et al., 2018; Cutkosky, 2019; Kavis et al., 2019; Rodomanov et al., 2024; Kreisler et al., 2024; Kovalev, 2025), as well as parameter-free variants (Cutkosky & Orabona, 2018; Orabona & Pál, 2021; Defazio & Mishchenko, 2023; Mishchenko & Defazio, 2023; Ivgi et al., 2023; Khaled et al., 2023; Kreisler et al., 2024), which do not require tuning the parameter $\eta$. Unfortunately, AdaGrad-type methods have a significant drawback: the stepsize $\eta_k$ in eq. (5) is non-increasing. Hence, it cannot truly adapt to the local curvature of the objective function, which may limit its performance in many applications (Defazio et al., 2022).

**Local curvature estimation.** In this paper, we focus on a different approach to computing the stepsize $\eta_k$ by estimating the local curvature, i.e., the local gradient Lipschitz constant. To the best of our knowledge, the first such algorithm that has strong convergence guarantees, GRAAL, was proposed by Malitsky (2020). It is a modification of GD, which uses the following stepsize rule at each iteration:

$$\eta_{k+1} = \min\Big\{(1+\gamma)\eta_k, \frac{\nu\lambda_{k+1}^2}{\eta_{k-1}}\Big\}, \tag{6}$$

where $\lambda_{k+1} > 0$ is a certain finite-difference estimate of the local inverse gradient Lipschitz constant at the current iteration, and $\gamma, \nu > 0$ are positive constants. Alacaoglu et al. (2023) showed that GRAAL can achieve the iteration complexity in eq. (3) for $L$-smooth functions. Moreover, Malitsky & Mishchenko (2020) established the same result for AdGD, the vanilla GD with a stepsize rule similar to eq. (6):

$$\eta_{k+1} = \min\Big\{\eta_k\sqrt{1 + \tfrac{\gamma\eta_k}{\eta_{k-1}}}, \nu\lambda_{k+1}\Big\}. \tag{7}$$

Overall, GRAAL and AdGD demonstrate attractive results, both theoretically and experimentally, on a range of practical optimization problems (Alacaoglu et al., 2023; Malitsky & Mishchenko, 2020).

## 1.3 Main Contribution: GRAAL with Nesterov Acceleration

Motivated by the attractive theoretical and practical results for GRAAL and AdGD, we pose the following natural research question:

**Q1.** *Is it possible to develop an algorithm that incorporates Nesterov acceleration and can truly adapt to the local curvature of the objective function, as GRAAL and AdGD do?*

Unfortunately, to the best of our knowledge, there is no positive and comprehensive answer to this question. Malitsky & Mishchenko (2020) proposed an accelerated version of AdGD, which showed strong experimental results. However, it is only a heuristic and does not have any theoretical convergence guarantees whatsoever. In addition, Li & Lan (2025) developed AC-FGM and Suh & Ma (2025) developed AdaNAG, which can be seen as attempts to incorporate Nesterov acceleration into GRAAL/AdGD with theoretical guarantees. However, the abilities of AC-FGM and AdaNAG to adapt to the local curvature of the objective function are highly limited, as we will discuss later.

In this paper, we provide a positive answer to Question 1 and make the following contributions:

(i) In Section 2, we develop Accelerated GRAAL (Algorithm 1) for solving problem (1), which incorporates Nesterov acceleration and utilizes a generalized version of the stepsize update rules in eqs. (6) and (7). We also provide a theoretical convergence analysis of this algorithm.

(ii) In Section 3, we show that Algorithm 1 achieves the optimal iteration complexity in eq. (4) for $L$-smooth functions up to additive logarithmic factors, without the requirement of hyperparameter tuning or any additional line search procedures.

(iii) In Section 4, we demonstrate the adaptive capabilities of Algorithm 1 by showing that it achieves the iteration complexity in eq. (4) under the more general $(L_0, L_1)$-smoothness of the objective function, up to constant additive factors that do not depend on the precision $\epsilon$.

The important feature of Algorithm 1 is that it can adapt the stepsize $\eta_k$ to the local curvature at a geometric, or linear, rate, just like the standard non-accelerated GRAAL. In contrast, AC-FGM and AdaNAG allow only sublinear growth of the stepsize, so their adaptive abilities are insufficient. In particular, Algorithm 1 is the first adaptive algorithm that can achieve near-optimal iteration complexity for $(L_0, L_1)$-smooth functions, while there are no such results for AC-FGM or AdaNAG, to the best of our knowledge. More details are available in Sections 3.2 and 4.2.

## 1.4 Related Work

**More adaptive stepsizes: Barzilai-Borwein and Polyak.** The idea of computing stepsizes using estimates of the local gradient Lipschitz constant was previously used by Barzilai & Borwein (1988), who proposed the following stepsize rule:

$$\eta_{k+1} = \frac{\langle x_{k+1} - x_k, \nabla f(x_{k+1}) - \nabla f(x_k) \rangle}{\|\nabla f(x_{k+1}) - \nabla f(x_k)\|^2}. \tag{8}$$

Unfortunately, GD with this rule provably works only in the case where the objective function $f(x)$ is quadratic (Raydan, 1993; Dai & Liao, 2002), and may not work otherwise (Burdakov et al., 2019). Polyak (1969) suggested using GD with the following stepsize rule:

$$\eta_{k+1} = \frac{f(x_{k+1}) - f(x^*)}{\|\nabla f(x_{k+1})\|^2}. \tag{9}$$

Similar to AdaGrad, GD with this rule was shown to be universal (Hazan & Kakade, 2019). However, it requires a tight estimate of the optimal objective function value $f(x^*)$, which is rarely available in practice.

**Optimization for $(L_0, L_1)$-smooth functions.** The $(L_0, L_1)$-smoothness assumption was proposed by Zhang et al. (2019) as a generalization and a more realistic alternative to the standard $L$-smoothness. The convergence of gradient methods under this assumption has been extensively studied in the literature (Zhang et al., 2020; Chen et al., 2023). Gorbunov et al. (2024) showed that AdGD can achieve the iteration complexity of non-accelerated GD in eq. (3) up to additive constant factors without the requirement of hyperparameter tuning or line search. Additionally, several accelerated algorithms with theoretical guarantees are available (Li et al., 2023; Gorbunov et al., 2024; Vankov et al., 2024). However, all these algorithms are non-adaptive, and only Vankov et al.

(2024) managed to achieve the optimal iteration complexity in eq. (4) up to additive constant factors, with the requirement of a substantially more complex small-dimensional relaxation oracle (Nesterov et al., 2021). It is also worth mentioning the concurrent work of Tyurin (2025), who managed to achieve the complexity in eq. (4) up to additive constant factors. However, their algorithm requires the tuning of several parameters and is, therefore, also non-adaptive. Additionally, the initial version of our paper appeared online prior to the work of Tyurin (2025), with only the results from Section 4 missing, which we were finalizing at that time.

## 2 ADAPTIVE GRADIENT METHOD WITH NESTEROV ACCELERATION

### 2.1 ALGORITHM DEVELOPMENT

In this section, we develop Accelerated GRAAL for solving problem (1). Below, we briefly describe the key ideas used in the development of the algorithm. After assembling these ideas, we obtain the resulting Algorithm 1.

**Local curvature estimator.** As discussed in Section 1.2, the stepsize rule in eq. (6) used in GRAAL requires an estimate of the inverse local gradient Lipschitz constant $\lambda$. When the objective function is convex, given two points $x, z \in \mathbb{R}^d$, we can consider the following two options for computing $\lambda$:

$$\text{Option I:} \quad \lambda = \frac{\|x-z\|}{\|\nabla f(x) - \nabla f(z)\|}, \qquad \text{Option II:} \quad \lambda = \frac{2\,\mathrm{D}_f(x,z)}{\|\nabla f(x) - \nabla f(z)\|^2}. \tag{10}$$

GRAAL was originally developed for solving monotone variational inequalities (VI). Hence, it uses Option I, which works for this more general problem class, with the gradient $\nabla f(x)$ replaced by the monotone operator. However, it turns out that Option II can better exploit the properties of the objective function $f(x)$. Hence, we use Option II and, for convenience, define $\Lambda(x; z)$ as follows:

$$\Lambda(x; z) = \begin{cases} \frac{2\,\mathrm{D}_f(x;z)}{\|\nabla f(x) - \nabla f(z)\|^2} & \nabla f(x) \neq \nabla f(z) \\ +\infty & \nabla f(x) = \nabla f(z) \end{cases}. \tag{11}$$

It is worth noting that Li & Lan (2025); Suh & Ma (2025) also used Option II in AC-FGM and AdaNAG, respectively.

**Nesterov acceleration.** To incorporate Nesterov acceleration in Algorithm 1, we use its recent interpretation by Kovalev & Borodich (2024). The idea is that at the iteration $k$ of a gradient method, we replace the objective function $f(x)$ with the function $f_k(x) \colon \mathbb{R}^d \to \mathbb{R}$, defined as follows:

$$f_k(x) = \alpha_k^{-1} \cdot f(\alpha_k x + (1 - \alpha_k)\overline{x}_k), \quad \text{where } \alpha_k \in (0, 1], \; \overline{x}_k \in \mathbb{R}^d. \tag{12}$$

In the case where GD is used, that is, $x_{k+1} = x_k - \eta_k \nabla f_k(x_k)$, we can choose $\alpha_k = 2/(k + 2)$ and $\overline{x}_{k+1} = \alpha_k x_{k+1} + (1 - \alpha_k)\overline{x}_k$, which gives the STM algorithm (Gasnikov & Nesterov, 2016), a variant of AGD. However, we use the definition in eq. (12) as it substantially simplifies the development of Algorithm 1.

**GRAAL extrapolation.** We use the extrapolation step of GRAAL in combination with the interpretation of Nesterov acceleration above. It can be summarized as follows:

$$x_{k+1} = x_k - \eta_k \nabla f_k(\hat{x}_k), \qquad \hat{x}_{k+1} = x_{k+1} + \theta(x_{k+1} - x_k), \tag{13}$$

where $\theta > 0$ is the extrapolation parameter. GRAAL uses extrapolation for two reasons. First, the vanilla gradient method does not work for VI as it diverges even on simple bilinear min-max problems (Daskalakis et al., 2017). Hence, the extragradient method (Korpelevich, 1976; Mishchenko et al., 2020) or methods with extrapolation (Daskalakis et al., 2017; Malitsky & Tam, 2020; Kovalev et al., 2022) are typically used. Second, and more importantly, to the best of our knowledge, the particular type of extrapolation used by GRAAL plays a key role in its adaptive capabilities. In particular, it is an open question whether our results can be obtained with a different baseline algorithm.

**Problem: choosing $\alpha_k$.** Although it may seem that the tools described above are already enough to obtain Algorithm 1, one issue remains. The interpretation of Kovalev & Borodich (2024) combined with the GRAAL step in eq. (13) suggests that one should choose $\overline{x}_{k+1} = \alpha_k \hat{x}_k + (1 - \alpha_k)\overline{x}_k$. However, this would require that the parameters $\alpha_k$ satisfy the following inequality:[1]

$$\eta_k/\alpha_k \leq \eta_{k-1}/\alpha_{k-1} + \eta_k. \tag{14}$$

---

[1]More details on eq. (14) are available in the works of Kovalev & Borodich (2024); Kovalev (2025).

---

**Algorithm 1** Accelerated GRAAL

---

1: **input:** $x_0 \in \mathbb{R}^d$, $\eta_0 > 0$, $K \in \{1, 2, \dots\}$
2: **parameters:** $\theta, \gamma, \nu > 0$ satisfying eq. (19)
3: $\alpha_0 = \beta_0 = 1$, $\quad H_0 = H_{-1} = \eta_{-1} = \eta_0$, $\quad \tilde{x}_0 = \overline{x}_0 = x_0$
4: **for** $k = 0, 1, \dots, K - 1$ **do**
5: $\quad \alpha_{k+1} = \frac{(1+\gamma)\eta_k}{H_k + (1+\gamma)\eta_k}$
6: $\quad x_{k+1} = x_k - \eta_k \nabla f(\tilde{x}_k)$          ▷ *gradient step*
7: $\quad \overline{x}_{k+1} = \beta_k \tilde{x}_k + (1 - \beta_k)\overline{x}_k$      ▷ *additional coupling step*
8: $\quad \hat{x}_{k+1} = x_{k+1} + \theta(x_{k+1} - x_k)$      ▷ *GRAAL extrapolation*
9: $\quad \tilde{x}_{k+1} = \alpha_{k+1}\hat{x}_{k+1} + (1 - \alpha_{k+1})\overline{x}_{k+1}$    ▷ *Nesterov acceleration/STM*
10: $\quad \lambda_{k+1} = \min\{\Lambda(\overline{x}_{k+1}; \tilde{x}_k), \Lambda(\overline{x}_{k+1}; \tilde{x}_{k+1})\}$    ▷ *local curvature estimator*
11: $\quad \eta_{k+1} = \min\left\{(1+\gamma)\eta_k, \frac{\nu H_{k-1}\lambda_{k+1}}{\eta_{k-1}}\right\}$, $H_{k+1} = H_k + \eta_{k+1}$    ▷ *adaptive stepsize*
12: $\quad \beta_{k+1} = \frac{\eta_{k+1}}{\alpha_{k+1}H_{k+1}}$
13: **output:** $\overline{x}_K \in \mathbb{R}^d$

---

The best option is to choose $\alpha_k$ such that eq. (14) becomes an equality. However, this is impossible: computing $\eta_k$ requires an estimate of the local curvature, which requires the computation of the gradient $\nabla f_k(\hat{x}_k)$ and its use in eq. (11), which in turn requires knowing $\alpha_k$ in advance. Alternatively, one could follow the approach of Li & Lan (2025) and Suh & Ma (2025) used in AC-FGM and AdaNAG, respectively, and simply predefine $\alpha_k \propto 2/(k+2)$, just like in AGD. However, this requires additional restrictions on the stepsize $\eta_k$ and vastly limits the adaptation capabilities of AC-FGM, as we will discuss in Section 3.2.

**Solution: additional coupling step.** The key idea to resolve the issue above is to avoid the inequality in eq. (14) by defining $\overline{x}_{k+1}$ differently, using an additional coupling step:

$$\overline{x}_{k+1} = \beta_k \tilde{x}_k + (1 - \beta_k)\overline{x}_k, \qquad \tilde{x}_k = \alpha_k \hat{x}_k + (1 - \alpha_k)\overline{x}_k, \tag{15}$$

where $\beta_k \in (0, 1]$. Consequently, instead of requiring the inequality eq. (14), we choose the parameters $\beta_k$ to satisfy the following relation:[2]

$$\eta_k/(\alpha_k\beta_k) = \eta_{k-1}/(\alpha_{k-1}\beta_{k-1}) + \eta_k = \cdots = H_k, \quad \text{where } H_k = \sum_{i=0}^{k}\eta_i. \tag{16}$$

Hence, we choose $\beta_k = \eta_k/(\alpha_k H_k)$ and avoid additional restrictions on the stepsize $\eta_k$. The only remaining question is how to ensure $\beta_k \leq 1$. The answer is that we choose $\alpha_k = \frac{(1+\gamma)\eta_{k-1}}{H_{k-1}+(1+\gamma)\eta_{k-1}}$. Indeed, in Lemma 1, we prove that $\beta_k \in (0, 1]$ by utilizing the inequality $\eta_k \leq (1+\gamma)\eta_{k-1}$, which is implied by our stepsize rule in eq. (17). Moreover, our choice of $\alpha_k$ is implementable as it does not require knowledge of $\eta_k$, and, in contrast to AC-FGM and AdaNAG, it is adaptive because it is not based on any predefined sequence, but rather uses the adaptive stepsizes $\eta_{k-1}$ and their sum $H_{k-1}$.

**Adaptive stepsize.** We use the following adaptive stepsize $\eta_k$ in our algorithm:

$$\eta_{k+1} = \min\left\{(1+\gamma)\eta_k, \frac{\nu H_{k-1}\lambda_{k+1}}{\eta_{k-1}}\right\}, \tag{17}$$

where $\lambda_{k+1} = \min\{\Lambda(\overline{x}_{k+1}; \tilde{x}_k), \Lambda(\overline{x}_{k+1}; \tilde{x}_{k+1})\}$ is the local curvature estimator. This rule is primarily implied by the convergence analysis in the proof of Theorem 1 in Appendix A.3. It can also be seen as a generalization of the stepsize rules in eqs. (6) and (7) for GRAAL and AdGD, respectively.

## 2.2 CONVERGENCE ANALYSIS

We start the convergence analysis with the following two lemmas. In Lemma 1, we show that $\beta_k \in (0, 1]$ as discussed in the previous Section 2.1. In Lemma 2, we use the additional coupling step from line 7 and the convexity of the objective function $f(x)$ to obtain some useful inequalities.

**Lemma 1** (↓). $\lambda_k, \eta_k, H_k > 0$, $\alpha_k \in (0, 1)$, *and* $\beta_k \in (0, 1]$ *for all* $k \in \{1, \dots, K\}$.

---

[2]More details on eq. (16) are available in the proof of Theorem 1 in Appendix A.3.

**Lemma 2** ($\downarrow$)**.** *The following inequalities hold for all* $k \in \{1, \dots, K-1\}$*:*
$$f(\overline{x}_k) - f(\tilde{x}_k) \leq \tfrac{1}{\beta_k}(f(\overline{x}_k) - f(\overline{x}_{k+1})), \quad \mathrm{D}_f(\overline{x}_k; \tilde{x}_{k-1}) \leq \mathrm{D}_f(\overline{x}_{k-1}; \tilde{x}_{k-1}). \quad (18)$$

Now, we obtain the main convergence result in Theorem 1. Note that Algorithm 1 requires the universal constant parameters $\theta, \gamma, \nu > 0$ to satisfy eq. (19). However, it is easy to verify that such parameters exist. Using Theorem 1, we also obtain the upper bound on the functional suboptimality and distance to the solution of problem (1) in Corollary 1.

**Theorem 1** ($\downarrow$)**.** *Let parameters* $\theta, \gamma, \nu > 0$ *satisfy the following relations:*
$$4\nu\theta(1+\gamma)^2 = \gamma, \quad 1 + 2\gamma + \tfrac{2\gamma\theta^2}{(1+\theta)^2} \leq \tfrac{\theta}{(1+\theta)} + \tfrac{\theta^2}{(1+\theta)^2}. \quad (19)$$

*Then, the following inequality holds for all* $x \in \mathbb{R}^d$ *and* $k \in \{1, \dots, K-1\}$*:*
$$\Psi_{k+1}(x) \leq \Psi_k(x) - \tfrac{\gamma\theta}{2}\eta_k^2\|\nabla f(\tilde{x}_k)\|^2 - \tfrac{1}{4(1+\gamma)}\eta_k\,\mathrm{D}_f(\overline{x}_k; \tilde{x}_k), \quad (20)$$

*where* $\Psi_k(x)$ *is defined as follows:*
$$\Psi_k(x) = \tfrac{1}{2}\|x_k - x\|^2 + H_{k-1}(f(\overline{x}_k) - f(x)) + \tfrac{\theta\eta_k\eta_{k-1}}{\lambda_k}\mathrm{D}_f(\overline{x}_{k-1}; \tilde{x}_{k-1}) + \tfrac{\gamma\theta}{2}\|x_k - x_{k-1}\|^2. \quad (21)$$

**Corollary 1** ($\downarrow$)**.** *The following inequality holds for all* $x \in \mathbb{R}^d$ *and* $K \geq 1$*:*
$$\tfrac{1}{2}\|x_K - x\|^2 + H_{K-1}(f(\overline{x}_K) - f(x)) \leq \tfrac{1}{2}\|x_0 - x\|^2 + \tfrac{(1+\gamma\theta)}{2}\eta_0^2\|\nabla f(x_0)\|^2. \quad (22)$$

It is important to highlight that the results in Theorem 1 and Corollary 1 are general and do not require any additional assumptions on the objective function other than convexity and continuous differentiability. Consequently, they do not imply any non-asymptotic convergence results for Algorithm 1. However, in Sections 3 and 4, we will establish particular iteration complexities in the cases where the objective function is $L$-smooth and $(L_0, L_1)$-smooth, respectively.

## 3 CONVERGENCE ANALYSIS FOR $L$-SMOOTH FUNCTIONS

### 3.1 MAIN RESULT

In this section, we establish the iteration complexity of Algorithm 1 in the case where the objective function $f(x)$ is $L$-smooth for $L > 0$. That is, the gradient $\nabla f(x)$ is $L$-Lipschitz:
$$\|\nabla f(x) - \nabla f(z)\| \leq L\|x - z\| \quad \text{for all} \ \ x, z \in \mathbb{R}^d. \quad (23)$$
We start with the following two lemmas. Lemma 3 provides a lower bound on the curvature estimates $\lambda_k$. This lemma is standard, and its proof is given, for instance, by Nesterov et al. (2018, Theorem 2.1.5). Lemma 4 bounds the growth of the cumulative sum $H_k$ of the stepsizes $\eta_k$.

**Lemma 3.** $\lambda_k \geq 1/L$ *for all* $k \in \{1, \dots, K\}$*.*

**Lemma 4** ($\downarrow$)**.** $H_{k-1} \leq H_k \leq (2+\gamma)H_{k-1}$ *for all* $k \in \{0, \dots, K\}$*.*

Now, we are ready to establish the lower bound on the cumulative sum $H_k$ of the stepsizes $\eta_k$ in the following Theorem 2. Using this bound, we establish the iteration complexity of Algorithm 1 for $L$-smooth functions in Corollary 2.

**Theorem 2** ($\downarrow$)**.** *Let constants* $c, m \in \mathbb{R}$ *be defined as follows:*
$$c = \min\left\{\tfrac{\sqrt{\nu}}{3(2+\gamma)}, \tfrac{\sqrt{\nu}\gamma}{16\sqrt[4]{\gamma(1+\gamma)^5(2+\gamma)^3}}\right\}, \quad m = \left\lceil\max\left\{2, \ln_{1+\gamma}\left[\tfrac{4c^2}{\gamma\eta_0 L}\right]\right\}\right\rceil. \quad (24)$$

*Then, the following inequality holds for all* $k \in \{0, \dots, K\}$*:*
$$\sqrt{H_k} \geq \tfrac{c}{\sqrt{L}} \cdot (k - m). \quad (25)$$

**Corollary 2.** *Let* $\eta_0 L \leq 1$*. Then, to reach the precision* $f(\overline{x}_K) - f(x^*) \leq \epsilon$*, the following number of iterations of Algorithm 1 is sufficient:*
$$K = \mathcal{O}\left(1 + \sqrt{L\|x_0 - x^*\|^2/\epsilon} + \ln\left[\tfrac{1}{\eta_0 L}\right]\right). \quad (26)$$

Note that the complexity result in Corollary 2 requires the initial stepsize $\eta_0$ to satisfy the inequality $\eta_0 \leq 1/L$. However, we can simply choose $\eta_0$ to be very small, say $10^{-10}$, as suggested by Malitsky & Mishchenko (2020) for AdGD. This will only result in a small logarithmic additive factor in the iteration complexity as implied by Corollary 2.

## 3.2 COMPARISON WITH AC-FGM AND ADANAG

**AC-FGM.** Li & Lan (2025, Corollary 1) use the following adaptive stepsize rule for AC-FGM:

$$\eta_{k+1} = \min\left\{\tfrac{k+1}{k}\eta_k, \tfrac{1}{8}k\lambda_{k+1}\right\}. \tag{27}$$

This rule implies that the stepsize growth is restricted by the inequality $\eta_{k+1} \leq (1 + 1/k)\eta_k$. As mentioned in Section 2.1, this restriction substantially limits the ability of AC-FGM to adapt to the curvature of the objective function. In particular, the result of Li & Lan (2025, Corollary 1) implies the following iteration complexity, provided $\eta_0 \leq 0.4/L$:

$$K = \mathcal{O}\left(\sqrt{\max\{1, 1/(\eta_0 L)\} \cdot L\|x_0 - x^*\|^2/\epsilon}\right) \quad \Rightarrow \quad f(\overline{x}_K) - f(x^*) \leq \epsilon. \tag{28}$$

This matches the optimal complexity in eq. (4), if we choose $\eta_0 = 0.4/L$. However, if we choose the initial stepsize to be too small, i.e., $\eta_0 \ll 0.4/L$, the complexity result in eq. (28) will be worse than the optimal one by a factor of $1/\sqrt{\eta_0 L}$. In other words, the stepsize rule in eq. (27) cannot adapt to a "bad" choice of the initial stepsize $\eta_0$ due to stepsize growth restrictions. Li & Lan (2025) even had to use a line search at the first iteration of AC-FGM to find a "good" initial stepsize $\eta_0$ and achieve the optimal complexity in eq. (4).

**AdaNAG.** Suh & Ma (2025) uses a stepsize rule in AdaNAG, which is not substantially different from eq. (27). Consequently, they encounter issues similar to AC-FGM. The difference is that they do not use a line search at the first iteration, but rather estimate $\eta_0$ using Option I in eq. (10), which implies the following iteration complexity:

$$K = \mathcal{O}\left(\max\{1, \eta_0 L\} \cdot \sqrt{L\|x_0 - x^*\|^2/\epsilon}\right) \quad \Rightarrow \quad f(\overline{x}_K) - f(x^*) \leq \epsilon. \tag{29}$$

This result may be significantly worse than the optimal one in eq. (4) if the initial stepsize estimate is too large, i.e., $\eta_0 L \gg 1$. Moreover, similar to AC-FGM, the growth of the stepsize in AdaNAG is also substantially restricted, which can limit its performance, for instance, under the more realistic $(L_0, L_1)$-smoothness assumption.

**Algorithm 1 vs AC-FGM and AdaNAG.** In contrast to AC-FGM and AdaNAG, our stepsize rule in eq. (17) allows the geometric growth of the stepsize $\eta_{k+1} \leq (1 + \gamma)\eta_k$. Hence, Algorithm 1 can adapt even to a very small choice of the initial stepsize $\eta_0$ at the cost of a small logarithmic additive factor in the iteration complexity, as indicated by Corollary 2. In addition, as we will discuss in Section 4, the geometric growth of the stepsize is crucial for adaptation under the $(L_0, L_1)$-smoothness assumption, where the local gradient Lipschitz constant may change at an exponential rate. It is also worth mentioning that Li & Lan (2025, Corollary 2) and Suh & Ma (2025, Theorem 6) tried to resolve the issues with the stepsize growth restrictions in AC-FGM and AdaNAG by using different stepsize rules. However, they could not properly justify the efficiency of these new stepsize rules and provably achieve geometric growth of the stepsize.

## 4 CONVERGENCE ANALYSIS FOR $(L_0, L_1)$-SMOOTH FUNCTIONS

In this section, we establish the iteration complexity of Algorithm 1 in the case where the objective function $f(x)$ is $(L_0, L_1)$-smooth for $L_0, L_1 > 0$. That is, the objective function $f(x)$ is twice continuously differentiable and the following inequality holds:

$$\|\nabla^2 f(x)\| \leq L_0 + L_1\|\nabla f(x)\| \quad \text{for all} \quad x \in \mathbb{R}^d. \tag{30}$$

This assumption was proposed by Zhang et al. (2019) and is primarily motivated by experiments suggesting that the norm of the Hessian correlates with the gradient norm of the objective functions in deep neural networks. Note that the requirement for twice continuous differentiability may be relaxed by using the following equivalent condition for continuously differentiable objective functions (Vankov et al., 2024, Lemma 2.5):

$$\|\nabla f(x) - \nabla f(z)\| \leq (L_0 + L_1\|\nabla f(x)\|) \cdot \tfrac{1}{L_1}(\exp(L_1\|x - z\|) - 1). \tag{31}$$

It is also important to highlight that $(L_0, L_1)$-smoothness implies $L$-smoothness with $L = L_0$. The reverse is obviously not true: $(L_0, L_1)$-smoothness is much more general and allows the exponential growth of the objective function and local gradient Lipschitz constant (Gorbunov et al., 2024, Lemma 2.1; Vankov et al., 2024, Lemma 2.5).

### 4.1 MAIN RESULT

We start the convergence analysis of Algorithm 1 with Lemma 5, which refines the previously obtained results in Theorem 1 and Corollary 1. Furthermore, in Lemmas 6 and 7, we establish lower bounds on the estimate $\lambda_k$ of the local inverse gradient Lipschitz constant.

**Lemma 5** ($\downarrow$). *The following inequalities hold for all $K \geq 1$:*

$$\|\tilde{x}_{K-1} - x^*\| \leq \mathcal{D}, \quad \|\overline{x}_{K-1} - x^*\| \leq \mathcal{D}, \quad \sum_{i=1}^{K}\left(\eta_i^2\|\nabla f(\tilde{x}_i)\|^2 + \eta_i\,\mathrm{D}_f(\overline{x}_i;\tilde{x}_i)\right) \leq \mathcal{D}^2, \quad (32)$$

*where $\mathcal{D} \geq 0$ is defined as follows:*

$$\mathcal{D}^2 = \max\left\{\frac{1}{\gamma\theta}, 2(1+\gamma), (1+2\theta)^2\right\}\left(\|x_0 - x^*\|^2 + (1+\gamma\theta)\eta_0^2\|\nabla f(x_0)\|^2\right). \quad (33)$$

**Lemma 6** ($\downarrow$). $\lambda_k \geq \lambda_{\min}$ *for all $k \in \{1, \ldots, K\}$, where $\lambda_{\min} > 0$ is defined as follows:*

$$\lambda_{\min} = \frac{1}{L_0}\exp(-3L_1\mathcal{D}). \quad (34)$$

**Lemma 7** ($\downarrow$). *The following inequality holds for all $k \in \{1, \ldots, K\}$:*

$$\lambda_k \geq \frac{4}{9\max\{2L_0, 2L_1\|\nabla f(\tilde{x}_k)\|, 2L_1\|\nabla f(\tilde{x}_{k-1})\|, 9L_1^2\,\mathrm{D}_f(\overline{x}_k, \tilde{x}_k), 9L_1^2\,\mathrm{D}_f(\overline{x}_{k-1}, \tilde{x}_{k-1})\}}. \quad (35)$$

Next, we define the sets of indices $\mathcal{T}_1(k), \mathcal{T}_2(k), \mathcal{T}_3(k), \mathcal{T}_4(k)$ as follows:

$$\begin{aligned}
\mathcal{T}_1(k) &= \left\{1 \leq i \leq k : \eta_i = \frac{\nu H_{i-2}\lambda_i}{\eta_{i-2}} \text{ and } \lambda_i \geq \frac{2}{9L_0}\right\}, \\
\mathcal{T}_2(k) &= \left\{1 \leq i \leq k : \eta_i = \frac{\nu H_{i-2}\lambda_i}{\eta_{i-2}} \text{ and } \lambda_i < \frac{2}{9L_0}\right\}, \\
\mathcal{T}_3(k) &= \left\{1 \leq i \leq k : \eta_i < \frac{\nu H_{i-2}\lambda_i}{\eta_{i-2}} \text{ and } (l(i) \in \mathcal{T}_1(k) \text{ or } i - l(i) > m)\right\}, \\
\mathcal{T}_4(k) &= \left\{1 \leq i \leq k : \eta_i < \frac{\nu H_{i-2}\lambda_i}{\eta_{i-2}} \text{ and } l(i) \in \mathcal{T}_2(k) \cup \{0\} \text{ and } i - l(i) \leq m\right\},
\end{aligned} \quad (36)$$

where $m$ is a positive integer, and the integer function $l(k)$ is defined as follows:

$$l(k) = \max\{i : i \in \mathcal{T}_1(k) \cup \mathcal{T}_2(k) \cup \{0\}\}. \quad (37)$$

It is not hard to verify that these sets of indices are pairwise disjoint and that $\cup_{j=1}^{4}\mathcal{T}_j(k) = \{1, \ldots, k\}$. Moreover, the sizes of the sets $\mathcal{T}_2(k)$ and $\mathcal{T}_4(k)$ are bounded as shown in the following Lemma 8.

**Lemma 8** ($\downarrow$). *The following inequalities hold for all $k \in \{1, \ldots, K\}$:*

$$|\mathcal{T}_4(k)| \leq m + m|\mathcal{T}_2(k)|, \qquad |\mathcal{T}_2(k)| \leq 1 + \frac{81(1+\gamma)^2}{2\min\{\nu, \nu^2\}} \cdot L_1^2\mathcal{D}^2. \quad (38)$$

Now, we establish the key lower bound on the cumulative sum $H_k$ of the stepsizes $\eta_k$ in the following Theorem 3. Using this bound, we establish the iteration complexity of Algorithm 1 for $(L_0, L_1)$-smooth functions in Corollary 3.

**Theorem 3** ($\downarrow$). *Let $c > 0$ and $m > 0$ be defined as follows:*

$$c = \min\left\{\frac{\sqrt{2\nu}}{9(2+\gamma)}, \frac{\sqrt{\nu}\gamma}{24\sqrt[4]{4\gamma(1+\gamma)^5(2+\gamma)^3}}\right\}, \quad m = \left\lceil\max\left\{1, \ln_{1+\gamma}\left[\frac{4c^2}{\gamma\eta_0 L_0}\right], 4\ln_{1+\gamma}\left[\frac{2}{9L_0\lambda_{\min}}\right]\right\}\right\rceil. \quad (39)$$

*Then, the following inequality holds for all $k \in \{0, \ldots, K\}$.*

$$\sqrt{H_k} \geq \frac{c}{\sqrt{L_0}} \cdot (k - |\mathcal{T}_2(k)| - |\mathcal{T}_4(k)| - 1). \quad (40)$$

**Corollary 3** ($\downarrow$). *Let $\eta_0 L_0 \exp(L_1\|x_0 - x^*\|) \leq 1$. Then, $\mathcal{D} = \mathcal{O}(\|x_0 - x^*\|)$, and to reach the precision $f(\overline{x}_K) - f(x^*) \leq \epsilon$, the following number of iterations of Algorithm 1 is sufficient:*

$$K = \mathcal{O}\left(1 + \sqrt{L_0\mathcal{D}^2/\epsilon} + L_1^3\mathcal{D}^3 + \left(1 + L_1^2\mathcal{D}^2\right)\ln\left[\frac{1}{\eta_0 L_0}\right]\right). \quad (41)$$

Similar to Corollary 2 for the $L$-smooth case, Corollary 3 requires the initial stepsize $\eta_0$ to satisfy the inequality $\eta_0 L_0 \exp(L_1\|x_0 - x^*\|) \leq 1$. We can ensure this inequality without any line search or hyperparameter tuning, simply by choosing a very small initial stepsize $\eta_0$. Choosing the initial stepsize $\eta_0$ too small will only result in an additive constant factor $\left(1 + L_1^2\mathcal{D}^2\right)\ln\left[\frac{1}{\eta_0 L_0}\right]$, which does not depend on the precision $\epsilon$ and has a logarithmic dependence on $\eta_0$.

Table 1: Comparison of the iteration complexities for solving problem (1) under the convexity and $(L_0, L_1)$-smoothness; universal constants are omitted; $\mathcal{D} = \|x_0 - x^*\|$; the initial functional gap is bounded as $f(x_0) - f(x^*) \leq \mathcal{O}(L_0\mathcal{D}^2 \exp(L_1\mathcal{D}))$, where necessary; optimality is considered up to additive constants.

| Reference | Iteration Complexity | Optimal | Adaptive |
|:---:|:---:|:---:|:---:|
| Li et al. (2023) | $\sqrt{\frac{L_0\mathcal{D}^2}{\epsilon}} \times \left(1 + L_1^2\mathcal{D}^2\right)\exp(\mathcal{O}(1)L_1\mathcal{D})$ | ✗ | ✗ |
| Gorbunov et al. (2024) | $\sqrt{\frac{L_0\mathcal{D}^2}{\epsilon}} \times \sqrt{1 + L_1\mathcal{D}\exp(L_1\mathcal{D})}$ | ✗ | ✗ |
| Vankov et al. (2024) | $\sqrt{\frac{L_0\mathcal{D}^2}{\epsilon}} + (L_1\mathcal{D})^{5/3}$ | ✔ | ✗ |
| Tyurin (2025) | $\sqrt{\frac{L_0\mathcal{D}^2}{\epsilon}} + (L_1\mathcal{D})^2$ | ✔ | ✗ |
| **Corollary 3** | $\sqrt{\frac{L_0\mathcal{D}^2}{\epsilon}} + (L_1\mathcal{D})^3$ | ✔ | ✔ |

## 4.2 COMPARISON WITH EXISTING RESULTS

**Accelerated methods for $(L_0, L_1)$-smooth functions.** As mentioned in Section 1.4, there are several existing accelerated algorithms with theoretical guarantees for minimizing convex $(L_0, L_1)$-smooth functions. To compare these results with Algorithm 1, we use a particular choice of the initial stepsize $\eta_0 = \frac{1}{L_0}\exp(-L_1\|x_0 - x^*\|)$ in Corollary 3. The comparison is summarized in Table 1. The algorithms of Li et al. (2023); Gorbunov et al. (2024) are neither adaptive nor optimal. The algorithms of Vankov et al. (2024); Tyurin (2025) are near-optimal as they match the complexity in eq. (4) up to additive constants, just like Algorithm 1. The result of Vankov et al. (2024) has a slightly better additive constant $(L_1\mathcal{D})^{5/3}$. However, neither of the algorithms of Vankov et al. (2024); Tyurin (2025) is adaptive: the algorithm of Vankov et al. (2024) requires solving a one-dimensional auxiliary optimization subproblem at each iteration, and the algorithm of Tyurin (2025) requires tuning several parameters. In contrast, Algorithm 1 does not require any hyperparameter tuning or line search to achieve near-optimal complexity, as discussed in the previous Section 4.1.

**AdGD for $(L_0, L_1)$-smooth functions.** Gorbunov et al. (2024) established the iteration complexity $\mathcal{O}\left(L_0\mathcal{D}^2/\epsilon + (L_1\mathcal{D})^6\right)$ for the AdGD algorithm under the $(L_0, L_1)$-smoothness assumption.[3] This result is unsurprisingly worse than ours in Corollary 3 due to the lack of acceleration. In addition, Gorbunov et al. (2024) did not prove that the constant $\mathcal{D}$ is bounded as $\mathcal{D} = \mathcal{O}(\|x_0 - x^*\|)$. In fact, $\mathcal{D}$ also contains the initial objective function gap $f(x_0) - f(x^*)$, and hence, it may have an exponential dependency on the initial distance $\|x_0 - x^*\|$.

**AC-FGM and AdaNAG.** As previously discussed in Section 3.2, we allow the geometric growth of the adaptive stepsize in Algorithm 1, which is crucial for obtaining the near-optimal complexity result in Corollary 3. Indeed, the estimates of the local curvature $\lambda_k$ can scale exponentially in the worst case according to Lemma 6, but can grow up to $\mathcal{O}(1/L_0)$ when the algorithm reaches a certain region near the solution $x^*$. Hence, the growth of the stepsize at a geometric rate or faster is necessary to avoid exponential factors in the iteration complexity. In contrast, Li & Lan (2025); Suh & Ma (2025) do not provide any convergence guarantees for $(L_0, L_1)$-smoothness for AC-FGM and AdaNAG. In addition, as previously discussed, the rate of stepsize growth in AC-FGM and AdaNAG is far below geometric. Hence, we conjecture that it is not possible to reach near-optimal complexity with these algorithms.

## ACKNOWLEDGEMENTS

This work was supported by the The Ministry of Economic Development of the Russian Federation in accordance with the subsidy agreement (agreement identifier 000000C313925P4G0002; grant No 139-15-2025-011).

---

[3]Note that Gorbunov et al. (2024) incorrectly reported their result as $\tilde{\mathcal{O}}\left(L_0\mathcal{D}^2/\epsilon + (L_1\mathcal{D})^4\right)$.

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

# Appendix

## A PROOFS FOR SECTION 2

### A.1 PROOF OF LEMMA 1

First, we can show that $\lambda_k > 0$ for all $k \in \{1, \ldots, K\}$. Otherwise, if $\lambda_k = 0$, there exist $x, z \in \mathbb{R}^d$ such that $\Lambda(x, z) = 0$ due to the definition of $\lambda_k$ on line 10. Hence, $D_f(x; z) = 0$ according to eq. (11). From the convexity of the function $f(x)$, it follows that $D_f(x; z) \geq 0$. Hence, $x$ minimizes the function $D_f(\cdot; z)$, and by the first-order optimality conditions, $\nabla_x D_f(x; z) = \nabla f(x) - \nabla f(z) = 0$. This implies $\Lambda(x; z) = +\infty$ according to eq. (11), which contradicts the possibility of $\lambda_k = 0$.

Next, using the fact that $\lambda_k > 0$ and lines 3 and 11, it is easy to show that $\eta_k, H_k > 0$ for all $k \in \{-1, \ldots, K\}$. Hence, the inclusions $\alpha_k \in (0, 1)$ and $\beta_k \in (0, +\infty)$ are obvious due to their definitions on lines 5 and 12. Finally, we can upper-bound $\beta_{k+1}$ as follows:

$$\beta_{k+1} \stackrel{(a)}{=} \frac{\eta_{k+1}}{\alpha_{k+1} H_{k+1}} \stackrel{(b)}{=} \frac{\eta_{k+1}}{H_k + \eta_{k+1}} \cdot \frac{H_k + (1+\gamma)\eta_k}{(1+\gamma)\eta_k} \stackrel{(c)}{\leq} \frac{(1+\gamma)\eta_k}{H_k + (1+\gamma)\eta_k} \cdot \frac{H_k + (1+\gamma)\eta_k}{(1+\gamma)\eta_k} = 1,$$

where (a) uses line 12; (b) uses lines 5 and 11; (c) uses the inequality $\eta_{k+1} \leq (1+\gamma)\eta_k$ implied by line 11 and the monotonicity of the function $t \mapsto t/(1+t)$. □

### A.2 PROOF OF LEMMA 2

We can upper-bound $f(\overline{x}_{k+1})$ as follows:

$$f(\overline{x}_{k+1}) \stackrel{(a)}{=} f(\beta_k \tilde{x}_k + (1-\beta_k)\overline{x}_k) \stackrel{(b)}{\leq} \beta_k f(\tilde{x}_k) + (1-\beta_k) f(\overline{x}_k),$$

where (a) uses line 7; (b) uses Lemma 1 and the convexity of $f(x)$. After rearranging, we obtain the first desired inequality. Furthermore, we can upper-bound $D_f(\overline{x}_k, \tilde{x}_{k-1})$ as follows:

$$
\begin{aligned}
D_f(\overline{x}_k; \tilde{x}_{k-1}) &\stackrel{(a)}{=} D_f(\beta_{k-1}\tilde{x}_{k-1} + (1-\beta_{k-1})\overline{x}_{k-1}; \tilde{x}_{k-1}) \\
&\stackrel{(b)}{\leq} (1-\beta_{k-1}) D_f(\overline{x}_{k-1}; \tilde{x}_{k-1}) + \beta_{k-1} D_f(\tilde{x}_{k-1}; \tilde{x}_{k-1}) \\
&\stackrel{(c)}{\leq} D_f(\overline{x}_{k-1}; \tilde{x}_{k-1})
\end{aligned}
$$

where (a) uses line 7; (b) and (c) use uses Lemma 1 and the convexity of $D_f(\cdot; \tilde{x}_{k-1})$, which is implied by the convexity of $f(x)$. This proves the second desired inequality. □

### A.3 PROOF OF THEOREM 1

For all $k \in \{1, \ldots, K-1\}$, we obtain the following:

$$
\begin{aligned}
\tfrac{1}{2}\|x_{k+1} - x\|^2 &= \tfrac{1}{2}\|x_k - x\|^2 - \tfrac{1}{2}\|x_{k+1} - x_k\|^2 + \langle x_{k+1} - x_k, x_{k+1} - x \rangle \\
&= \tfrac{1}{2}\|x_k - x\|^2 - \tfrac{1}{2}\|x_{k+1} - x_k\|^2 + \langle x_{k+1} - x_k, x_{k+1} - \hat{x}_{k+1} \rangle \\
&\quad + \langle x_{k+1} - x_k, \hat{x}_{k+1} - \hat{x}_k + \hat{x}_k - x \rangle \\
&\stackrel{(a)}{=} \tfrac{1}{2}\|x_k - x\|^2 - \left(\tfrac{1}{2} + \theta\right)\|x_{k+1} - x_k\|^2 + \langle x_{k+1} - x_k, \hat{x}_{k+1} - \hat{x}_k + \hat{x}_k - x \rangle \\
&\stackrel{(b)}{=} \tfrac{1}{2}\|x_k - x\|^2 - \left(\tfrac{1}{2} + \theta\right)\|x_{k+1} - x_k\|^2 - \eta_k \langle \nabla f(\tilde{x}_k), \hat{x}_{k+1} - \hat{x}_k \rangle \\
&\quad + \eta_k \langle \nabla f(\tilde{x}_k), x - \hat{x}_k \rangle \\
&\stackrel{(c)}{=} \tfrac{1}{2}\|x_k - x\|^2 - \left(\tfrac{1}{2} + \theta\right)\|x_{k+1} - x_k\|^2 - \eta_k \langle \nabla f(\tilde{x}_k), \hat{x}_{k+1} - \hat{x}_k \rangle \\
&\quad + \eta_k \langle \nabla f(\tilde{x}_k), x - \tilde{x}_k \rangle + \tfrac{(1-\alpha_k)\eta_k}{\alpha_k} \langle \nabla f(\tilde{x}_k), \overline{x}_k - \tilde{x}_k \rangle \\
&\stackrel{(d)}{\leq} \tfrac{1}{2}\|x_k - x\|^2 - \left(\tfrac{1}{2} + \theta\right)\|x_{k+1} - x_k\|^2 - \eta_k \langle \nabla f(\tilde{x}_k), \hat{x}_{k+1} - \hat{x}_k \rangle
\end{aligned}
$$

$$+ \eta_k(f(x) - f(\tilde{x}_k)) + \tfrac{(1-\alpha_k)\eta_k}{\alpha_k}(f(\overline{x}_k) - f(\tilde{x}_k) - D_f(\overline{x}_k; \tilde{x}_k))$$

$$= \tfrac{1}{2}\|x_k - x\|^2 - \left(\tfrac{1}{2} + \theta\right)\|x_{k+1} - x_k\|^2 - \eta_k\langle\nabla f(\tilde{x}_k), \hat{x}_{k+1} - \hat{x}_k\rangle$$

$$- \eta_k(f(\overline{x}_k) - f(x)) + \tfrac{\eta_k}{\alpha_k}(f(\overline{x}_k) - f(\tilde{x}_k)) - \tfrac{(1-\alpha_k)\eta_k}{\alpha_k} D_f(\overline{x}_k; \tilde{x}_k)$$

$$\overset{(e)}{\leq} \tfrac{1}{2}\|x_k - x\|^2 - \left(\tfrac{1}{2} + \theta\right)\|x_{k+1} - x_k\|^2 - \eta_k\langle\nabla f(\tilde{x}_k), \hat{x}_{k+1} - \hat{x}_k\rangle$$

$$- \eta_k(f(\overline{x}_k) - f(x)) + \tfrac{\eta_k}{\alpha_k\beta_k}(f(\overline{x}_k) - f(\overline{x}_{k+1})) - \tfrac{(1-\alpha_k)\eta_k}{\alpha_k} D_f(\overline{x}_k; \tilde{x}_k)$$

$$\overset{(f)}{=} \tfrac{1}{2}\|x_k - x\|^2 - \left(\tfrac{1}{2} + \theta\right)\|x_{k+1} - x_k\|^2 - \eta_k\langle\nabla f(\tilde{x}_k), \hat{x}_{k+1} - \hat{x}_k\rangle$$

$$- \eta_k(f(\overline{x}_k) - f(x)) + H_k(f(\overline{x}_k) - f(\overline{x}_{k+1})) - \tfrac{(1-\alpha_k)\eta_k}{\alpha_k} D_f(\overline{x}_k; \tilde{x}_k)$$

$$\overset{(g)}{=} \tfrac{1}{2}\|x_k - x\|^2 - \left(\tfrac{1}{2} + \theta\right)\|x_{k+1} - x_k\|^2 - \eta_k\langle\nabla f(\tilde{x}_k), \hat{x}_{k+1} - \hat{x}_k\rangle$$

$$+ H_{k-1}(f(\overline{x}_k) - f(x)) - H_k(f(\overline{x}_{k+1}) - f(x)) - \tfrac{(1-\alpha_k)\eta_k}{\alpha_k} D_f(\overline{x}_k; \tilde{x}_k),$$

where (a) uses line 8; (b) uses line 6; (c) uses line 9; (d) uses the convexity of function $f(x)$; (e) uses Lemma 2; (f) uses line 12; (g) uses line 11. Furthermore, we get the following:

$$\tfrac{1}{2}\|x_{k+1} - x\|^2$$

$$\leq \tfrac{1}{2}\|x_k - x\|^2 - \left(\tfrac{1}{2} + \theta\right)\|x_{k+1} - x_k\|^2 + H_{k-1}(f(\overline{x}_k) - f(x)) - H_k(f(\overline{x}_{k+1}) - f(x))$$

$$- \tfrac{(1-\alpha_k)\eta_k}{\alpha_k} D_f(\overline{x}_k; \tilde{x}_k) - \eta_k\langle\nabla f(\tilde{x}_k) - \nabla f(\tilde{x}_{k-1}), \hat{x}_{k+1} - \hat{x}_k\rangle - \eta_k\langle\nabla f(\tilde{x}_{k-1}), \hat{x}_{k+1} - \hat{x}_k\rangle$$

$$\overset{(a)}{=} \tfrac{1}{2}\|x_k - x\|^2 - \left(\tfrac{1}{2} + \theta\right)\|x_{k+1} - x_k\|^2 + H_{k-1}(f(\overline{x}_k) - f(x)) - H_k(f(\overline{x}_{k+1}) - f(x))$$

$$- \tfrac{(1-\alpha_k)\eta_k}{\alpha_k} D_f(\overline{x}_k; \tilde{x}_k) - \eta_k\langle\nabla f(\tilde{x}_k) - \nabla f(\tilde{x}_{k-1}), \hat{x}_{k+1} - \hat{x}_k\rangle$$

$$+ \tfrac{\eta_k}{\eta_{k-1}}\langle x_k - x_{k-1}, \hat{x}_{k+1} - \hat{x}_k\rangle$$

$$\overset{(b)}{=} \tfrac{1}{2}\|x_k - x\|^2 - \left(\tfrac{1}{2} + \theta\right)\|x_{k+1} - x_k\|^2 + H_{k-1}(f(\overline{x}_k) - f(x)) - H_k(f(\overline{x}_{k+1}) - f(x))$$

$$- \tfrac{(1-\alpha_k)\eta_k}{\alpha_k} D_f(\overline{x}_k; \tilde{x}_k) - \eta_k\langle\nabla f(\tilde{x}_k) - \nabla f(\tilde{x}_{k-1}), \hat{x}_{k+1} - \hat{x}_k\rangle$$

$$+ \tfrac{\eta_k}{\theta\eta_{k-1}}\langle\hat{x}_k - x_k, \hat{x}_{k+1} - \hat{x}_k\rangle$$

$$= \tfrac{1}{2}\|x_k - x\|^2 - \left(\tfrac{1}{2} + \theta\right)\|x_{k+1} - x_k\|^2 + H_{k-1}(f(\overline{x}_k) - f(x)) - H_k(f(\overline{x}_{k+1}) - f(x))$$

$$- \tfrac{(1-\alpha_k)\eta_k}{\alpha_k} D_f(\overline{x}_k; \tilde{x}_k) - \eta_k\langle\nabla f(\tilde{x}_k) - \nabla f(\tilde{x}_{k-1}), \hat{x}_{k+1} - \hat{x}_k\rangle$$

$$+ \tfrac{\eta_k}{2\theta\eta_{k-1}}\left(\|\hat{x}_{k+1} - x_k\|^2 - \|\hat{x}_k - x_k\|^2 - \|\hat{x}_{k+1} - \hat{x}_k\|^2\right)$$

$$\overset{(c)}{=} \tfrac{1}{2}\|x_k - x\|^2 - \left(\tfrac{1}{2} + \theta\right)\|x_{k+1} - x_k\|^2 + H_{k-1}(f(\overline{x}_k) - f(x)) - H_k(f(\overline{x}_{k+1}) - f(x))$$

$$- \tfrac{(1-\alpha_k)\eta_k}{\alpha_k} D_f(\overline{x}_k; \tilde{x}_k) - \eta_k\langle\nabla f(\tilde{x}_k) - \nabla f(\tilde{x}_{k-1}), \hat{x}_{k+1} - \hat{x}_k\rangle$$

$$+ \tfrac{\eta_k}{2\theta\eta_{k-1}}\left((1+\theta)^2\|x_{k+1} - x_k\|^2 - \theta^2\|x_k - x_{k-1}\|^2 - \|\hat{x}_{k+1} - \hat{x}_k\|^2\right)$$

$$\overset{(d)}{\leq} \tfrac{1}{2}\|x_k - x\|^2 + \tfrac{(1+\theta)^2}{2\theta}\left(\tfrac{\eta_k}{\eta_{k-1}} - \tfrac{\theta+2\theta^2}{(1+\theta)^2}\right)\|x_{k+1} - x_k\|^2 - \tfrac{\theta\eta_k}{2\eta_{k-1}}\|x_k - x_{k-1}\|^2$$

$$- \tfrac{\eta_k}{2\theta\eta_{k-1}}\|\hat{x}_{k+1} - \hat{x}_k\|^2 + \eta_k\|\nabla f(\tilde{x}_k) - \nabla f(\tilde{x}_{k-1})\|\|\hat{x}_{k+1} - \hat{x}_k\|$$

$$- \tfrac{(1-\alpha_k)\eta_k}{\alpha_k} D_f(\overline{x}_k; \tilde{x}_k) + H_{k-1}(f(\overline{x}_k) - f(x)) - H_k(f(\overline{x}_{k+1}) - f(x)),$$

where (a) uses line 6; (b) and (c) use line 8; (d) uses the Cauchy-Schwarz inequality. Next, we get

$$\tfrac{1}{2}\|x_{k+1} - x\|^2$$

$$\leq \tfrac{1}{2}\|x_k - x\|^2 + \tfrac{(1+\theta)^2}{2\theta}\left(\tfrac{\eta_k}{\eta_{k-1}} - \tfrac{\theta+2\theta^2}{(1+\theta)^2}\right)\|x_{k+1} - x_k\|^2 - \tfrac{\theta\eta_k}{2\eta_{k-1}}\|x_k - x_{k-1}\|^2$$

$$- \tfrac{(1-\alpha_k)\eta_k}{\alpha_k} D_f(\overline{x}_k; \tilde{x}_k) + H_{k-1}(f(\overline{x}_k) - f(x)) - H_k(f(\overline{x}_{k+1}) - f(x))$$

$$- \tfrac{\eta_k}{2\theta\eta_{k-1}}\|\hat{x}_{k+1} - \hat{x}_k\|^2 + \eta_k\|\nabla f(\tilde{x}_k) - \nabla f(\tilde{x}_{k-1})\|\|\hat{x}_{k+1} - \hat{x}_k\|$$

$$\overset{(a)}{\leq} \tfrac{1}{2}\|x_k - x\|^2 + H_{k-1}(f(\overline{x}_k) - f(x)) - H_k(f(\overline{x}_{k+1}) - f(x))$$

$$+ \tfrac{(1+\theta)^2}{2\theta}\left(\tfrac{\eta_k}{\eta_{k-1}} - \tfrac{\theta+2\theta^2}{(1+\theta)^2}\right)\|x_{k+1} - x_k\|^2 - \tfrac{\theta\eta_k}{2\eta_{k-1}}\|x_k - x_{k-1}\|^2$$

$$- \frac{\eta_k}{2\theta\eta_{k-1}}\|\hat{x}_{k+1} - \hat{x}_k\|^2 - \frac{(1-\alpha_k)\eta_k}{\alpha_k}\mathrm{D}_f(\overline{x}_k; \tilde{x}_k)$$

$$+ \eta_k\|\nabla f(\tilde{x}_k) - \nabla f(\overline{x}_k)\|\|\hat{x}_{k+1} - \hat{x}_k\| + \eta_k\|\nabla f(\overline{x}_k) - \nabla f(\tilde{x}_{k-1})\|\|\hat{x}_{k+1} - \hat{x}_k\|$$

$$\overset{(b)}{\leq} \frac{1}{2}\|x_k - x\|^2 + H_{k-1}(f(\overline{x}_k) - f(x)) - H_k(f(\overline{x}_{k+1}) - f(x))$$

$$+ \frac{(1+\theta)^2}{2\theta}\left(\frac{\eta_k}{\eta_{k-1}} - \frac{\theta+2\theta^2}{(1+\theta)^2}\right)\|x_{k+1} - x_k\|^2 - \frac{\theta\eta_k}{2\eta_{k-1}}\|x_k - x_{k-1}\|^2$$

$$- \frac{\eta_k}{2\theta\eta_{k-1}}\|\hat{x}_{k+1} - \hat{x}_k\|^2 - \frac{(1-\alpha_k)\eta_k}{\alpha_k}\mathrm{D}_f(\overline{x}_k; \tilde{x}_k)$$

$$+ \eta_k\sqrt{\frac{2}{\lambda_k}\mathrm{D}_f(\overline{x}_k; \tilde{x}_k)}\|\hat{x}_{k+1} - \hat{x}_k\| + \eta_k\sqrt{\frac{2}{\lambda_k}\mathrm{D}_f(\overline{x}_k; \tilde{x}_{k-1})}\|\hat{x}_{k+1} - \hat{x}_k\|$$

$$\overset{(c)}{\leq} \frac{1}{2}\|x_k - x\|^2 + H_{k-1}(f(\overline{x}_k) - f(x)) - H_k(f(\overline{x}_{k+1}) - f(x))$$

$$+ \frac{(1+\theta)^2}{2\theta}\left(\frac{\eta_k}{\eta_{k-1}} - \frac{\theta+2\theta^2}{(1+\theta)^2}\right)\|x_{k+1} - x_k\|^2 - \frac{\theta\eta_k}{2\eta_{k-1}}\|x_k - x_{k-1}\|^2$$

$$- \frac{\eta_k}{2\theta\eta_{k-1}}\|\hat{x}_{k+1} - \hat{x}_k\|^2 - \frac{(1-\alpha_k)\eta_k}{\alpha_k}\mathrm{D}_f(\overline{x}_k; \tilde{x}_k)$$

$$+ \eta_k\sqrt{\frac{2}{\lambda_k}\mathrm{D}_f(\overline{x}_k; \tilde{x}_k)}\|\hat{x}_{k+1} - \hat{x}_k\| + \eta_k\sqrt{\frac{2}{\lambda_k}\mathrm{D}_f(\overline{x}_{k-1}; \tilde{x}_{k-1})}\|\hat{x}_{k+1} - \hat{x}_k\|$$

$$\overset{(d)}{\leq} \frac{1}{2}\|x_k - x\|^2 + H_{k-1}(f(\overline{x}_k) - f(x)) - H_k(f(\overline{x}_{k+1}) - f(x))$$

$$+ \frac{(1+\theta)^2}{2\theta}\left(\frac{\eta_k}{\eta_{k-1}} - \frac{\theta+2\theta^2}{(1+\theta)^2}\right)\|x_{k+1} - x_k\|^2 - \frac{\theta\eta_k}{2\eta_{k-1}}\|x_k - x_{k-1}\|^2$$

$$- \frac{\eta_k}{2\theta\eta_{k-1}}\|\hat{x}_{k+1} - \hat{x}_k\|^2 - \frac{(1-\alpha_k)\eta_k}{\alpha_k}\mathrm{D}_f(\overline{x}_k; \tilde{x}_k) + \frac{(1-\alpha_k)\eta_k}{2\alpha_k}\mathrm{D}_f(\overline{x}_k; \tilde{x}_k)$$

$$+ \frac{\alpha_k\eta_k}{(1-\alpha_k)\lambda_k}\|\hat{x}_{k+1} - \hat{x}_k\|^2 + \frac{\theta\eta_k\eta_{k-1}}{\lambda_k}\mathrm{D}_f(\overline{x}_{k-1}; \tilde{x}_{k-1}) + \frac{\eta_k}{2\theta\eta_{k-1}}\|\hat{x}_{k+1} - \hat{x}_k\|^2$$

$$= \frac{1}{2}\|x_k - x\|^2 + H_{k-1}(f(\overline{x}_k) - f(x)) - H_k(f(\overline{x}_{k+1}) - f(x)) - \frac{(1-\alpha_k)\eta_k}{4\alpha_k}\mathrm{D}_f(\overline{x}_k; \tilde{x}_k)$$

$$+ \frac{(1+\theta)^2}{2\theta}\left(\frac{\eta_k}{\eta_{k-1}} - \frac{\theta+2\theta^2}{(1+\theta)^2}\right)\|x_{k+1} - x_k\|^2 - \frac{\theta\eta_k}{2\eta_{k-1}}\|x_k - x_{k-1}\|^2$$

$$+ \frac{\alpha_k\eta_k}{(1-\alpha_k)\lambda_k}\|\hat{x}_{k+1} - \hat{x}_k\|^2 - \frac{(1-\alpha_k)\eta_k}{4\alpha_k}\mathrm{D}_f(\overline{x}_k; \tilde{x}_k) + \frac{\theta\eta_k\eta_{k-1}}{\lambda_k}\mathrm{D}_f(\overline{x}_{k-1}; \tilde{x}_{k-1})$$

$$\overset{(e)}{\leq} \frac{1}{2}\|x_k - x\|^2 + H_{k-1}(f(\overline{x}_k) - f(x)) - H_k(f(\overline{x}_{k+1}) - f(x)) - \frac{\eta_k}{4(1+\gamma)}\mathrm{D}_f(\overline{x}_k; \tilde{x}_k)$$

$$+ \frac{(1+\theta)^2}{2\theta}\left(\frac{\eta_k}{\eta_{k-1}} - \frac{\theta+2\theta^2}{(1+\theta)^2}\right)\|x_{k+1} - x_k\|^2 - \frac{\theta\eta_k}{2\eta_{k-1}}\|x_k - x_{k-1}\|^2$$

$$+ \frac{\alpha_k\eta_k}{(1-\alpha_k)\lambda_k}\|\hat{x}_{k+1} - \hat{x}_k\|^2 - \frac{(1-\alpha_k)\eta_k}{4\alpha_k}\mathrm{D}_f(\overline{x}_k; \tilde{x}_k) + \frac{\theta\eta_k\eta_{k-1}}{\lambda_k}\mathrm{D}_f(\overline{x}_{k-1}; \tilde{x}_{k-1})$$

where (a) uses the triangle inequality; (b) uses line 10 and eq. (11); (c) uses Lemma 2; (d) uses Young's inequality; (e) uses lines 5 and 11. Furthermore, we obtain the following:

$$\frac{1}{2}\|x_{k+1} - x\|^2$$

$$\leq \frac{1}{2}\|x_k - x\|^2 + H_{k-1}(f(\overline{x}_k) - f(x)) - H_k(f(\overline{x}_{k+1}) - f(x)) - \frac{\eta_k}{4(1+\gamma)}\mathrm{D}_f(\overline{x}_k; \tilde{x}_k)$$

$$+ \frac{(1+\theta)^2}{2\theta}\left(\frac{\eta_k}{\eta_{k-1}} - \frac{\theta+2\theta^2}{(1+\theta)^2}\right)\|x_{k+1} - x_k\|^2 - \frac{\theta\eta_k}{2\eta_{k-1}}\|x_k - x_{k-1}\|^2$$

$$+ \frac{\alpha_k\eta_k}{(1-\alpha_k)\lambda_k}\|\hat{x}_{k+1} - \hat{x}_k\|^2 - \frac{(1-\alpha_k)\lambda_{k+1}}{4\theta\alpha_k\eta_{k+1}} \cdot \frac{\theta\eta_{k+1}\eta_k}{\lambda_{k+1}}\mathrm{D}_f(\overline{x}_k; \tilde{x}_k) + \frac{\theta\eta_k\eta_{k-1}}{\lambda_k}\mathrm{D}_f(\overline{x}_{k-1}; \tilde{x}_{k-1})$$

$$\overset{(a)}{\leq} \frac{1}{2}\|x_k - x\|^2 + H_{k-1}(f(\overline{x}_k) - f(x)) - H_k(f(\overline{x}_{k+1}) - f(x)) - \frac{\eta_k}{4(1+\gamma)}\mathrm{D}_f(\overline{x}_k; \tilde{x}_k)$$

$$+ \frac{(1+\theta)^2}{2\theta}\left(\frac{\eta_k}{\eta_{k-1}} - \frac{\theta+2\theta^2}{(1+\theta)^2}\right)\|x_{k+1} - x_k\|^2 - \frac{\theta\eta_k}{2\eta_{k-1}}\|x_k - x_{k-1}\|^2$$

$$+ \frac{\alpha_k\eta_k}{(1-\alpha_k)\lambda_k}\|\hat{x}_{k+1} - \hat{x}_k\|^2 - \frac{1}{4\nu\theta(1+\gamma)} \cdot \frac{\theta\eta_{k+1}\eta_k}{\lambda_{k+1}}\mathrm{D}_f(\overline{x}_k; \tilde{x}_k) + \frac{\theta\eta_k\eta_{k-1}}{\lambda_k}\mathrm{D}_f(\overline{x}_{k-1}; \tilde{x}_{k-1})$$

$$\overset{(b)}{=} \frac{1}{2}\|x_k - x\|^2 + H_{k-1}(f(\overline{x}_k) - f(x)) - H_k(f(\overline{x}_{k+1}) - f(x)) - \frac{\eta_k}{4(1+\gamma)}\mathrm{D}_f(\overline{x}_k; \tilde{x}_k)$$

$$+ \frac{(1+\theta)^2}{2\theta}\left(\frac{\eta_k}{\eta_{k-1}} - \frac{\theta+2\theta^2}{(1+\theta)^2}\right)\|x_{k+1} - x_k\|^2 - \frac{\theta\eta_k}{2\eta_{k-1}}\|x_k - x_{k-1}\|^2$$

$$+ \frac{\alpha_k\eta_k}{(1-\alpha_k)\lambda_k}\|\hat{x}_{k+1} - \hat{x}_k\|^2 - \frac{(1+\gamma)}{\gamma} \cdot \frac{\theta\eta_{k+1}\eta_k}{\lambda_{k+1}}\mathrm{D}_f(\overline{x}_k; \tilde{x}_k) + \frac{\theta\eta_k\eta_{k-1}}{\lambda_k}\mathrm{D}_f(\overline{x}_{k-1}; \tilde{x}_{k-1})$$

$$\leq \frac{1}{2}\|x_k - x\|^2 + H_{k-1}(f(\overline{x}_k) - f(x)) - H_k(f(\overline{x}_{k+1}) - f(x)) - \frac{\eta_k}{4(1+\gamma)}\mathrm{D}_f(\overline{x}_k; \tilde{x}_k)$$

$$+ \frac{(1+\theta)^2}{2\theta}\left(\frac{\eta_k}{\eta_{k-1}} - \frac{\theta+2\theta^2}{(1+\theta)^2}\right)\|x_{k+1} - x_k\|^2 - \frac{\theta\eta_k}{2\eta_{k-1}}\|x_k - x_{k-1}\|^2$$

$$+ \frac{\alpha_k\eta_k}{(1-\alpha_k)\lambda_k}\|\hat{x}_{k+1} - \hat{x}_k\|^2 - \frac{\theta\eta_{k+1}\eta_k}{\lambda_{k+1}}\mathrm{D}_f(\overline{x}_k;\tilde{x}_k) + \frac{\theta\eta_k\eta_{k-1}}{\lambda_k}\mathrm{D}_f(\overline{x}_{k-1};\tilde{x}_{k-1})$$

$$\overset{(c)}{\leq} \frac{1}{2}\|x_k - x\|^2 + H_{k-1}(f(\overline{x}_k) - f(x)) - H_k(f(\overline{x}_{k+1}) - f(x)) - \frac{\eta_k}{4(1+\gamma)}\mathrm{D}_f(\overline{x}_k;\tilde{x}_k)$$

$$+ \frac{\theta\eta_k\eta_{k-1}}{\lambda_k}\mathrm{D}_f(\overline{x}_{k-1};\tilde{x}_{k-1}) - \frac{\theta\eta_{k+1}\eta_k}{\lambda_{k+1}}\mathrm{D}_f(\overline{x}_k;\tilde{x}_k)$$

$$+ \frac{(1+\theta)^2}{2\theta}\left(\frac{\eta_k}{\eta_{k-1}} - \frac{\theta+2\theta^2}{(1+\theta)^2}\right)\|x_{k+1} - x_k\|^2 - \frac{\theta\eta_k}{2\eta_{k-1}}\|x_k - x_{k-1}\|^2$$

$$+ \frac{2(1+\theta)^2\alpha_k\eta_k}{(1-\alpha_k)\lambda_k}\|x_{k+1} - x_k\|^2 + \frac{2\theta^2\alpha_k\eta_k}{(1-\alpha_k)\lambda_k}\|x_k - x_{k-1}\|^2$$

$$\leq \frac{1}{2}\|x_k - x\|^2 + H_{k-1}(f(\overline{x}_k) - f(x)) - H_k(f(\overline{x}_{k+1}) - f(x)) - \frac{\eta_k}{4(1+\gamma)}\mathrm{D}_f(\overline{x}_k;\tilde{x}_k)$$

$$+ \frac{\theta\eta_k\eta_{k-1}}{\lambda_k}\mathrm{D}_f(\overline{x}_{k-1};\tilde{x}_{k-1}) - \frac{\theta\eta_{k+1}\eta_k}{\lambda_{k+1}}\mathrm{D}_f(\overline{x}_k;\tilde{x}_k)$$

$$+ \frac{(1+\theta)^2}{2\theta}\left(\frac{\eta_k}{\eta_{k-1}} + \frac{4\theta\alpha_k\eta_k}{(1-\alpha_k)\lambda_k} - \frac{\theta+2\theta^2}{(1+\theta)^2}\right)\|x_{k+1} - x_k\|^2 + \frac{2\theta^2\alpha_k\eta_k}{(1-\alpha_k)\lambda_k}\|x_k - x_{k-1}\|^2$$

$$\overset{(d)}{\leq} \frac{1}{2}\|x_k - x\|^2 + H_{k-1}(f(\overline{x}_k) - f(x)) - H_k(f(\overline{x}_{k+1}) - f(x)) - \frac{\eta_k}{4(1+\gamma)}\mathrm{D}_f(\overline{x}_k;\tilde{x}_k)$$

$$+ \frac{\theta\eta_k\eta_{k-1}}{\lambda_k}\mathrm{D}_f(\overline{x}_{k-1};\tilde{x}_{k-1}) - \frac{\theta\eta_{k+1}\eta_k}{\lambda_{k+1}}\mathrm{D}_f(\overline{x}_k;\tilde{x}_k)$$

$$+ \frac{(1+\theta)^2}{2\theta}\left((1+\gamma) + \frac{4\theta\alpha_k\eta_k}{(1-\alpha_k)\lambda_k} - \frac{\theta+2\theta^2}{(1+\theta)^2}\right)\|x_{k+1} - x_k\|^2 + \frac{2\theta^2\alpha_k\eta_k}{(1-\alpha_k)\lambda_k}\|x_k - x_{k-1}\|^2$$

$$\overset{(e)}{\leq} \frac{1}{2}\|x_k - x\|^2 + H_{k-1}(f(\overline{x}_k) - f(x)) - H_k(f(\overline{x}_{k+1}) - f(x)) - \frac{\eta_k}{4(1+\gamma)}\mathrm{D}_f(\overline{x}_k;\tilde{x}_k)$$

$$+ \frac{\theta\eta_k\eta_{k-1}}{\lambda_k}\mathrm{D}_f(\overline{x}_{k-1};\tilde{x}_{k-1}) - \frac{\theta\eta_{k+1}\eta_k}{\lambda_{k+1}}\mathrm{D}_f(\overline{x}_k;\tilde{x}_k)$$

$$+ \frac{(1+\theta)^2}{2\theta}\left((1+\gamma) + 4\nu\theta(1+\gamma)^2 - \frac{\theta+2\theta^2}{(1+\theta)^2}\right)\|x_{k+1} - x_k\|^2 + 2\nu\theta^2(1+\gamma)^2\|x_k - x_{k-1}\|^2$$

$$\overset{(f)}{=} \frac{1}{2}\|x_k - x\|^2 + H_{k-1}(f(\overline{x}_k) - f(x)) - H_k(f(\overline{x}_{k+1}) - f(x)) - \frac{\eta_k}{4(1+\gamma)}\mathrm{D}_f(\overline{x}_k;\tilde{x}_k)$$

$$+ \frac{\theta\eta_k\eta_{k-1}}{\lambda_k}\mathrm{D}_f(\overline{x}_{k-1};\tilde{x}_{k-1}) - \frac{\theta\eta_{k+1}\eta_k}{\lambda_{k+1}}\mathrm{D}_f(\overline{x}_k;\tilde{x}_k)$$

$$+ \frac{(1+\theta)^2}{2\theta}\left(1 + 2\gamma - \frac{\theta}{(1+\theta)} - \frac{\theta^2}{(1+\theta)^2}\right)\|x_{k+1} - x_k\|^2 + \frac{\gamma\theta}{2}\|x_k - x_{k-1}\|^2$$

$$\overset{(g)}{\leq} \frac{1}{2}\|x_k - x\|^2 + H_{k-1}(f(\overline{x}_k) - f(x)) - H_k(f(\overline{x}_{k+1}) - f(x)) - \frac{\eta_k}{4(1+\gamma)}\mathrm{D}_f(\overline{x}_k;\tilde{x}_k)$$

$$+ \frac{\theta\eta_k\eta_{k-1}}{\lambda_k}\mathrm{D}_f(\overline{x}_{k-1};\tilde{x}_{k-1}) - \frac{\theta\eta_{k+1}\eta_k}{\lambda_{k+1}}\mathrm{D}_f(\overline{x}_k;\tilde{x}_k) - \gamma\theta\|x_{k+1} - x_k\|^2 + \frac{\gamma\theta}{2}\|x_k - x_{k-1}\|^2$$

$$\overset{(h)}{=} \frac{1}{2}\|x_k - x\|^2 + H_{k-1}(f(\overline{x}_k) - f(x)) - H_k(f(\overline{x}_{k+1}) - f(x))$$

$$+ \frac{\theta\eta_k\eta_{k-1}}{\lambda_k}\mathrm{D}_f(\overline{x}_{k-1};\tilde{x}_{k-1}) - \frac{\theta\eta_{k+1}\eta_k}{\lambda_{k+1}}\mathrm{D}_f(\overline{x}_k;\tilde{x}_k) - \frac{\gamma\theta}{2}\|x_{k+1} - x_k\|^2 + \frac{\gamma\theta}{2}\|x_k - x_{k-1}\|^2$$

$$- \frac{\gamma\theta\eta_k^2}{2}\|\nabla f(\tilde{x}_k)\|^2 - \frac{\eta_k}{4(1+\gamma)}\mathrm{D}_f(\overline{x}_k;\tilde{x}_k)m$$

where (a) and (e) use lines 5 and 11; (b), (f) and (g) use eq. (19); (c) uses line 8 and the inequality $\|a + b\|^2 \leq 2\|a\|^2 + 2\|b\|^2$; (d) uses the inequality $\eta_k \leq (1 + \gamma)\eta_{k-1}$ implied by line 11; (h) desc. It remains to use the definition $\Psi_k(x)$ in eq. (21). □

## A.4 Proof of Corollary 1

We can upper-bound $\frac{1}{2}\|x_K - x\|^2 + H_{K-1}(f(\overline{x}_K) - f(x))$ as follows:

$$\frac{1}{2}\|x_K - x\|^2 + H_{K-1}(f(\overline{x}_K) - f(x))$$

$$\overset{(a)}{\leq} \Psi_K(x) \overset{(b)}{\leq} \Psi_1(x)$$

$$\overset{(c)}{\leq} \frac{1}{2}\|x_1 - x\|^2 + H_0(f(\overline{x}_1) - f(x)) + \frac{\theta\eta_k\eta_{k-1}}{\lambda_k}\mathrm{D}_f(\overline{x}_0;\tilde{x}_0) + \frac{\gamma\theta}{2}\|x_1 - x_0\|^2$$

$$\overset{(d)}{=} \frac{1}{2}\|x_1 - x\|^2 + \eta_0(f(x_0) - f(x)) + \frac{\gamma\theta}{2}\|x_1 - x_0\|^2$$

$$\overset{(e)}{=} \tfrac{1}{2}\|x_0 - x\|^2 + \tfrac{(1+\gamma\theta)\eta_0^2}{2}\|\nabla f(x_0)\|^2 - \eta_0\langle\nabla f(x_0), x_0 - x\rangle + \eta_0(f(x_0) - f(x))$$

$$= \tfrac{1}{2}\|x_0 - x\|^2 + \tfrac{(1+\gamma\theta)\eta_0^2}{2}\|\nabla f(x_0)\|^2 - \eta_0\, \mathrm{D}_f(x, x_0)$$

$$\overset{(f)}{\leq} \tfrac{1}{2}\|x_0 - x\|^2 + \tfrac{(1+\gamma\theta)\eta_0^2}{2}\|\nabla f(x_0)\|^2,$$

where (a) and (c) use eq. (21); (b) uses Theorem 1; (d) uses lines 3 and 7; (e) use lines 3 and 6; (f) uses the convexity of $f(x)$. $\qquad\square$

## B  PROOFS FOR SECTION 3

### B.1  PROOF OF LEMMA 4

For $k = 0$ we have $H_k = H_{k-1}$. The inequality $H_k \geq H_{k-1}$ is obvious for $k \geq 1$ due to line 11. Furthermore, we can upper-bound $H_k$ for $k \geq 1$ as follows:

$$H_k \overset{(a)}{=} H_{k-1} + \eta_k \overset{(b)}{\leq} H_{k-1} + (1+\gamma)\eta_{k-1} \overset{(c)}{\leq} (2+\gamma)H_{k-1},$$

where (a), (b) and (c) use line 11.  □

### B.2  PROOF OF THEOREM 2

We prove the statement of Theorem 2 by induction over $K$. The base case, $K = m$, is obvious. Now, we assume that eq. (25) holds for all $k \in \{0, \ldots, K\}$, where $K \geq m$, and prove eq. (25) for $k = K + 1$. We consider the following cases:

**Case 1.** $(1+\gamma)\eta_K \eta_{K-1} \geq \nu H_{K-1} \lambda_{K+1}$.
**Case 2.** $(1+\gamma)\eta_K \eta_{K-1} < \nu H_{K-1} \lambda_{K+1}$.
    **Case 2a.** $\eta_K = \eta_0 (1+\gamma)^K$.
    **Case 2b.** $\eta_K < \eta_0 (1+\gamma)^K$.

**Case 1.** Here we have $\eta_{K+1} = \frac{\nu H_{K-1} \lambda_{k+1}}{\eta_{K-1}}$ and obtain the following inequality:

$$\sqrt{H_{K+1}} = \frac{1}{\sqrt{H_{K+1}} + \sqrt{H_{K-2}}}(H_{K+1} - H_{K-2}) + \sqrt{H_{K-2}}$$

$$\overset{(a)}{\geq} \frac{1}{2\sqrt{H_{K+1}}}(H_{K+1} - H_{K-2}) + \sqrt{H_{K-2}}$$

$$\overset{(b)}{\geq} \frac{1}{2\sqrt{H_{K+1}}}(\eta_{K+1} + \eta_{K-1}) + \sqrt{H_{K-2}}$$

$$\overset{(c)}{\geq} \sqrt{\frac{\eta_{K+1}\eta_{K-1}}{H_{K+1}}} + \sqrt{H_{K-2}}$$

$$\overset{(d)}{\geq} \sqrt{\frac{\nu H_{K-1}}{L H_{K+1}}} + \sqrt{H_{K-2}}$$

$$\overset{(e)}{\geq} \frac{\sqrt{\nu}}{(2+\gamma)\sqrt{L}} + \sqrt{H_{K-2}}$$

$$\overset{(f)}{\geq} \frac{\sqrt{\nu}}{(2+\gamma)\sqrt{L}} + \frac{c}{\sqrt{L}} \cdot (K - 2 - m)$$

$$\overset{(g)}{\geq} \frac{c}{\sqrt{L}} \cdot (K + 1 - m),$$

where (a) uses the inequality $H_{K-2} \leq H_{K+1}$ for $K \geq m \geq 2$; (b) uses line 11; (c) uses Young's inequality; (d) uses the assumption $\eta_{K+1} = \frac{\nu H_{K-1}\lambda_{k+1}}{\eta_{K-1}}$ (Case 1) and Lemma 3; (e) uses Lemma 4; (f) uses the induction hypothesis in eq. (25); (g) uses the inequality $\frac{3c}{\sqrt{L}} \leq \frac{\sqrt{\nu}}{(2+\gamma)\sqrt{L}}$ implied by eq. (24). This proves Case 1.

**Case 2a.** In this case, it is easy to verify that $\eta_k = \eta_0(1+\gamma)^k$ for all $k \in \{0, \ldots, K+1\}$. Hence, we can lower-bound $H_{K+1}$ as follows:

$$H_{K+1} \overset{(a)}{=} \sum_{k=0}^{K+1} \eta_0(1+\gamma)^k = \frac{\eta_0}{\gamma}\left((1+\gamma)^{K+2} - 1\right)$$

$$= \frac{\eta_0(1+\gamma)^m}{\gamma}\left((1+\gamma)^{K+2-m} - \frac{1}{(1+\gamma)^m}\right) \geq \frac{\eta_0(1+\gamma)^m}{\gamma}\left(\left((1+\gamma)^{(K-m+2)/2}\right)^2 - 1\right)$$

$$\overset{(b)}{\geq} \frac{\eta_0(1+\gamma)^m}{\gamma}\left(\left(1 + \frac{\gamma}{2}(K - m + 2)\right)^2 - 1\right) \geq \frac{\eta_0\gamma(1+\gamma)^m}{4}(K - m + 2)^2 \qquad (42)$$

$$\overset{(c)}{\geq} \frac{c^2}{L}(K + 2 - m)^2 \geq \frac{c^2}{L}(K + 1 - m)^2,$$

where (a) uses line 11 and the relation $\eta_k = \eta_0(1+\gamma)^k$ above; (b) uses the inequality $K \geq m$ and Bernoulli's inequality; (c) uses the inequality $(1+\gamma)^m \geq \frac{4c^2}{\gamma \eta_0 L}$ implied by eq. (24). This proves Case 2a.

**Case 2b.** In this case, there exists $l \in \{1, ..., K\}$ such that $\eta_l < (1+\gamma)\eta_{l-1}$. We choose the largest such index $l \in \{1, ..., K\}$. This implies the following relations:

$$\eta_l = \frac{\nu H_{l-2}\lambda_l}{\eta_{l-2}} \quad \text{and} \quad \eta_k = (1+\gamma)^{k-l}\eta_l \quad \text{for all} \ \ k \in \{l, \ldots, K+1\}. \tag{43}$$

Hence, we can upper-bound $H_{K+1}$ as follows:

$$H_{K+1} \overset{(a)}{=} H_{l-1} + \sum_{i=0}^{K-l+1} \eta_l(1+\gamma)^i = H_{l-1} + \frac{\eta_l}{\gamma}((1+\gamma)^{K-l+2} - 1)$$
$$\overset{(b)}{\leq} \left(1 + \frac{(1+\gamma)^{K-l+3} - (1+\gamma)}{\gamma}\right)H_{l-1} \leq \frac{(1+\gamma)^{K-l+3}}{\gamma}H_{l-1} \tag{44}$$

where (a) uses the above relation; (b) uses the inequalities $\eta_l \leq (1+\gamma)\eta_{l-1}$ and $\eta_{l-1} \leq H_{l-1}$ implied by line 11. Furthermore, we can lower-bound $\eta_l$ as follows:

$$\eta_l \overset{(a)}{=} \frac{\nu H_{l-2}\lambda_l}{\eta_{l-2}} \overset{(b)}{\geq} \frac{\nu H_{l-2}\lambda_l}{H_l - H_{l-3}} \overset{(c)}{\geq} \frac{\nu H_{l-2}}{L(H_l - H_{l-3})}, \tag{45}$$

where (a) uses the relation above; (b) uses lines 3 and 11, where we define $H_{-2} = H_0$; (c) uses Lemma 3. Next, we obtain the following inequality:

$$\left(\sqrt{H_{K+1}} - \sqrt{H_{l-3}}\right)^2 \overset{(a)}{\geq} \left(\sqrt{H_{K+1}} - \sqrt{H_{l-1}}\right)\left(\sqrt{H_l} - \sqrt{H_{l-3}}\right)$$
$$= \frac{H_{K+1} - H_{l-1}}{\sqrt{H_{K+1}} + \sqrt{H_{l-1}}} \cdot \frac{H_l - H_{l-3}}{\sqrt{H_l} + \sqrt{H_{l-3}}}$$
$$\overset{(b)}{\geq} \frac{H_{K+1} - H_{l-1}}{2\sqrt{H_{K+1}}} \cdot \frac{H_l - H_{l-3}}{2\sqrt{H_l}}$$
$$\overset{(c)}{=} \frac{\eta_l((1+\gamma)^{K-l+2} - 1)}{2\gamma\sqrt{H_{K+1}}} \cdot \frac{H_l - H_{l-3}}{2\sqrt{H_l}}$$
$$\overset{(d)}{\geq} \frac{\nu H_{l-2}((1+\gamma)^{K-l+2} - 1)}{4\gamma L\sqrt{H_{K+1}H_l}}$$
$$\overset{(e)}{\geq} \frac{\nu H_{l-2}}{4L\sqrt{\gamma(1+\gamma)H_{l-1}H_l}} \cdot \frac{(1+\gamma)^{K-l+2} - 1}{\sqrt{(1+\gamma)^{K-l+2}}}$$
$$\geq \frac{\nu H_{l-2}}{4L\sqrt{\gamma(1+\gamma)H_{l-1}H_l}} \cdot ((1+\gamma)^{(K-l)/2+1} - 1)$$
$$\overset{(f)}{\geq} \frac{\nu}{4L\sqrt{\gamma(1+\gamma)(2+\gamma)^3}} \cdot ((1+\gamma)^{(K-l)/2+1} - 1), \tag{46}$$

where (a) uses the inequalities $H_{l-3} \leq H_{l-1}$ and $H_l \leq H_{K+1}$; (b) uses the inequalities $H_{K+1} \geq H_{l-1}$ and $H_l \geq H_{l-3}$; (c) and (e) use eq. (44); (d) use eq. (45); (f) uses Lemma 4. Next, we take the square root of both sides and obtain the following:

$$\sqrt{H_{K+1}} - \sqrt{H_{l-3}} \geq \frac{\sqrt{\nu}}{2\sqrt[4]{\gamma(1+\gamma)(2+\gamma)^3 L^2}} \cdot \sqrt{(1+\gamma)^{(K-l)/2+1} - 1}$$
$$\overset{(a)}{\geq} \frac{\sqrt{\nu}}{2\sqrt[4]{\gamma(1+\gamma)(2+\gamma)^3 L^2}} \cdot \left((1+\gamma)^{(K-l+2)/4} - 1\right)$$
$$= \frac{\sqrt{\nu}}{2\sqrt[4]{\gamma(1+\gamma)^5(2+\gamma)^3 L^2}} \cdot \left((1+\gamma)^{(K-l)/4+3/2} - (1+\gamma)\right)$$
$$\overset{(b)}{\geq} \frac{\sqrt{\nu\gamma}}{2\sqrt[4]{\gamma(1+\gamma)^5(2+\gamma)^3 L^2}} \cdot \left(\frac{1}{4}(K-l) + \frac{1}{2}\right)$$
$$\overset{(c)}{\geq} \frac{\sqrt{\nu\gamma}}{2\sqrt[4]{\gamma(1+\gamma)^5(2+\gamma)^3 L^2}} \cdot \left(\frac{1}{8}(K-l) + \frac{1}{2}\right)$$
$$= \frac{\sqrt{\nu\gamma}}{16\sqrt[4]{\gamma(1+\gamma)^5(2+\gamma)^3 L^2}} \cdot (K-l+4) \tag{47}$$
$$\overset{(d)}{\geq} \frac{c}{\sqrt{L}}(K-l+4),$$

where (a) uses the inequality $\sqrt{a} \geq \sqrt{a+b} - \sqrt{b}$ for $a, b \geq 0$; (b) uses Bernoulli's inequality; (c) uses the inequality $K \geq l$; (d) uses eq. (24). Finally, we obtain the following:

$$\sqrt{H_{K+1}} \geq \sqrt{H_{l-3}} + \tfrac{c}{\sqrt{L}}(K - l + 4) \overset{(a)}{\geq} \tfrac{c}{\sqrt{L}}(K + 1 - m),$$

where (a) uses the induction hypothesis in eq. (25) for $k = l - 3$, which also holds for $l = 1$ due to the definition $H_{-2} = H_0$. This proves Case 2b. $\qquad\square$

# C  PROOFS FOR SECTION 4

## C.1  PROOF OF LEMMA 5

**Step 1.** We can upper-bound $\sum_{i=1}^{K}\left(\eta_i^2\|\nabla f(\tilde{x}_i)\|^2 + \eta_i\, \mathrm{D}_f(\overline{x}_i;\tilde{x}_i)\right)$ as follows:

$$\sum_{i=1}^{K}\left(\eta_i^2\|\nabla f(\tilde{x}_i)\|^2 + \eta_i\, \mathrm{D}_f(\overline{x}_i;\tilde{x}_i)\right)$$
$$\leq \max\left\{\tfrac{2}{\gamma\theta}, 4(1+\gamma)\right\}\sum_{i=1}^{K}\left(\tfrac{\gamma\theta}{2}\eta_i^2\|\nabla f(\tilde{x}_i)\|^2 + \tfrac{1}{4(1+\gamma)}\eta_i\, \mathrm{D}_f(\overline{x}_i;\tilde{x}_i)\right)$$
$$\overset{(a)}{\leq} \max\left\{\tfrac{2}{\gamma\theta}, 4(1+\gamma)\right\}\sum_{i=1}^{K}\left(\Psi_i(x^*) - \Psi_{i+1}(x^*)\right)$$
$$\overset{(b)}{\leq} \max\left\{\tfrac{2}{\gamma\theta}, 4(1+\gamma)\right\}\Psi_1(x^*)$$
$$\overset{(c)}{\leq} \max\left\{\tfrac{1}{\gamma\theta}, 2(1+\gamma)\right\}\left(\|x_0 - x^*\|^2 + (1+\gamma\theta)\eta_0^2\|\nabla f(x_0)\|^2\right),$$

where (a) uses Theorem 1; (b) uses the fact that $\Psi_1(x^*) \geq 0$, which is implied by eq. (21); (c) is obtained similarly to the proof of Corollary 1 in Appendix A.4.

**Step 2.** Next, we prove the following inequality for all $k \in \{0, \ldots, K\}$ by induction:

$$\max\{\|\tilde{x}_k - x^*\|, \|\overline{x}_k - x^*\|\} \leq (1+2\theta)\max_{k=0,\ldots,K}\|x_k - x^*\|. \tag{48}$$

The base case $k = 0$ is obvious due to line 3. For $k \geq 1$ we can upper-bound $\|\overline{x}_k - x^*\|$ as follows:

$$\|\overline{x}_k - x^*\| \overset{(a)}{=} \beta_{k-1}\|\tilde{x}_{k-1} - x^*\| + (1-\beta_{k-1})\|\overline{x}_{k-1} - x^*\|$$
$$\overset{(b)}{\leq} (1+2\theta)\max_{k=0,\ldots,K}\|x_k - x^*\|,$$

where (a) uses line 7, the triangle inequality, and Lemma 1; (b) uses the induction hypothesis in eq. (48) for $k - 1$. Next, we can upper-bound $\|\tilde{x}_k - x^*\|$ as follows:

$$\|\tilde{x}_k - x^*\| \overset{(a)}{\leq} \alpha_k\|\hat{x}_k - x^*\| + (1-\alpha_k)\|\overline{x}_k - x^*\|$$
$$\overset{(b)}{\leq} \alpha_k\|\hat{x}_k - x^*\| + (1-\alpha_k)(1+2\theta)\max_{k=0,\ldots,K}\|x_k - x^*\|$$
$$\overset{(c)}{\leq} \alpha_k(1+\theta)\|x_k - x^*\| + \alpha_k\theta\|x_{k-1} - x^*\| + (1-\alpha_k)(1+2\theta)\max_{k=0,\ldots,K}\|x_k - x^*\|$$
$$\leq (1+2\theta)\max_{k=0,\ldots,K}\|x_k - x^*\|,$$

where (a) uses line 9 and the triangle inequality; (b) uses the inequality obtained above; (c) uses line 8 and the triangle inequality. This proves eq. (48). Next, using eq. (48) and Corollary 1, we obtain the following inequality for $k \in \{0, \ldots, K\}$:

$$\max\left\{\|\tilde{x}_k - x^*\|^2, \|\overline{x}_k - x^*\|^2\right\} \leq (1+2\theta)^2\left(\|x_0 - x^*\|^2 + (1+\gamma\theta)\eta_0^2\|\nabla f(x_0)\|^2\right).$$

Combining this with the inequality obtained in Step 1 concludes the proof.  □

## C.2  PROOF OF LEMMA 6

We can lower-bound $\lambda_k$ as follows:

$$\lambda_k \overset{(a)}{=} \min\{\Lambda(\overline{x}_k, \tilde{x}_{k-1}), \Lambda(\overline{x}_k, \tilde{x}_k)\}$$
$$\overset{(b)}{=} \min\left\{\frac{2\,\mathrm{D}_f(\overline{x}_k,\tilde{x}_{k-1})}{\|\nabla f(\overline{x}_k) - \nabla f(\tilde{x}_{k-1})\|^2}, \frac{2\,\mathrm{D}_f(\overline{x}_k,\tilde{x}_k)}{\|\nabla f(\overline{x}_k) - \nabla f(\tilde{x}_k)\|^2}\right\}$$
$$\overset{(c)}{\geq} \frac{2}{2(L_0 + L_1\|\nabla f(\overline{x}_k)\|) + L_1\max\{\|\nabla f(\overline{x}_k) - \nabla f(\tilde{x}_k)\|, \|\nabla f(\overline{x}_k) - \nabla f(\tilde{x}_{k-1})\|\}}$$
$$\overset{(d)}{\geq} \frac{2}{(L_0 + L_1\|\nabla f(\overline{x}_k)\|)(\exp(L_1\max\{\|\overline{x}_k - \tilde{x}_k\|, \|\overline{x}_k - \tilde{x}_{k-1}\|\}) + 1)}$$

$$\geq \frac{2}{(L_0 + L_1\|\nabla f(\overline{x}_k)\|)(\exp(L_1(\|\overline{x}_k - x^*\| + \max\{\|\tilde{x}_k - x^*\|, \|\tilde{x}_{k-1} - x^*\|\})) + 1)}$$

$$\overset{(e)}{\geq} \frac{2}{(L_0 + L_1\|\nabla f(\overline{x}_k)\|)(1 + \exp(2L_1\mathcal{D}))}$$

$$\overset{(f)}{\geq} \frac{2}{L_0\exp(L_1\|\overline{x}_k - x^*\|)(1 + \exp(2L_1\mathcal{D}))}$$

$$\overset{(g)}{\geq} \frac{2}{L_0\exp(L_1\mathcal{D})(1 + \exp(2L_1\mathcal{D}))} \geq \frac{1}{L_0\exp(3L_1\mathcal{D})},$$

where (a) uses line 10; (b) uses eq. (11); (c) uses Corollary 2.8 of Vankov et al. (2024); (d) and (f) use eq. (31); (e) and (g) use Lemma 5. $\qquad \square$

### C.3 PROOF OF LEMMA 7

We can lower-bound $\mathrm{D}_f(x, z)$ for all $x, z \in \mathbb{R}^d$ as follows:

$$\mathrm{D}_f(x, z) \overset{(a)}{\geq} \frac{\|\nabla f(x) - \nabla f(z)\|^2}{2(L_0 + L_1\|\nabla f(x)\|) + L_1\|\nabla f(x) - \nabla f(z)\|}$$

$$\overset{(b)}{\geq} \frac{\|\nabla f(x) - \nabla f(z)\|^2}{2(L_0 + L_1\|\nabla f(z)\|) + 3L_1\|\nabla f(x) - \nabla f(z)\|},$$

where (a) uses Corollary 2.8 of Vankov et al. (2024); (b) uses the triangle inequality. Hence, we obtain the following inequality, which is quadratic in $\|\nabla f(x) - \nabla f(z)\|$:

$$\|\nabla f(x) - \nabla f(z)\|^2 - 3L_1\,\mathrm{D}_f(x, z) \cdot \|\nabla f(x) - \nabla f(z)\| - 2(L_0 + L_1\|\nabla f(z)\|)\,\mathrm{D}_f(x, z) \leq 0.$$

Solving this inequality with respect to $\|\nabla f(x) - \nabla f(z)\|$ gives the following

$$\|\nabla f(x) - \nabla f(z)\| \leq \frac{3L_1\,\mathrm{D}_f(x, z) + \sqrt{(3L_1\,\mathrm{D}_f(x, z))^2 + 8(L_0 + L_1\|\nabla f(z)\|)\,\mathrm{D}_f(x, z)}}{2}$$

$$\overset{(a)}{\leq} 3L_1\,\mathrm{D}_f(x, z) + \sqrt{2(L_0 + L_1\|\nabla f(z)\|)\,\mathrm{D}_f(x, z)}$$

$$\overset{(b)}{\leq} \tfrac{9}{2}L_1\,\mathrm{D}_f(x, z) + \tfrac{1}{3}(L_0/L_1 + \|\nabla f(z)\|),$$

where (a) uses the inequality $\sqrt{a + b} \leq \sqrt{a} + \sqrt{b}$; (b) uses Young's inequality. Combining this with the inequalities above gives the following:

$$\Lambda(x, z) \overset{(a)}{\equiv} \frac{2\,\mathrm{D}_f(x, z)}{\|\nabla f(x) - \nabla f(z)\|^2}$$

$$\overset{(b)}{\geq} \frac{2}{2(L_0 + L_1\|\nabla f(z)\|) + 3L_1\|\nabla f(x) - \nabla f(z)\|}$$

$$\overset{(c)}{\geq} \frac{4}{6(L_0 + L_1\|\nabla f(z)\|) + 27L_1^2\,\mathrm{D}_f(x, z)}$$

$$\geq \frac{4}{9\max\{2L_0, 2L_1\|\nabla f(z)\|, 9L_1^2\,\mathrm{D}_f(x, z)\}}$$

where (a) uses eq. (11); (b) and (c) use the inequalities obtained above. Using this, we can lower-bound $\lambda_k$ as follows:

$$\lambda_k \overset{(a)}{\geq} \min\{\Lambda(\overline{x}_k, \tilde{x}_{k-1}), \Lambda(\overline{x}_k, \tilde{x}_k)\}$$

$$\overset{(b)}{\geq} \frac{4}{9\max\{2L_0, 2L_1\|\nabla f(\tilde{x}_k)\|, 2L_1\|\nabla f(\tilde{x}_{k-1})\|, 9L_1^2\,\mathrm{D}_f(\overline{x}_k, \tilde{x}_k), 9L_1^2\,\mathrm{D}_f(\overline{x}_k, \tilde{x}_{k-1})\}}$$

$$\overset{(c)}{\geq} \frac{4}{9\max\{2L_0, 2L_1\|\nabla f(\tilde{x}_k)\|, 2L_1\|\nabla f(\tilde{x}_{k-1})\|, 9L_1^2\,\mathrm{D}_f(\overline{x}_k, \tilde{x}_k), 9L_1^2\,\mathrm{D}_f(\overline{x}_{k-1}, \tilde{x}_{k-1})\}},$$

where (a) uses line 10; (b) uses the inequality obtained above; (c) uses Lemma 2. $\qquad \square$

## C.4 PROOF OF LEMMA 8

Using the definition of $\mathcal{T}_4(k)$ in eq. (36), one can verify that the following inclusion holds:

$$\mathcal{T}_4(k) \subset \{1 \leq i \leq k : i - \max\{j : j \in \mathcal{T}_2(i) \cup \{0\}\} \leq m\}, \tag{49}$$

which implies the first desired inequality. Next, using Lemma 7 and the definition of $\mathcal{T}_2(k)$ in eq. (36), one can veryfy that the following inequality holds for all $i \in \mathcal{T}_2(k)$:

$$\lambda_i \geq \frac{4}{9\max\{2L_1\|\nabla f(\tilde{x}_i)\|, 2L_1\|\nabla f(\tilde{x}_{i-1})\|, 9L_1^2\,\mathrm{D}_f(\overline{x}_i, \tilde{x}_i), 9L_1^2\,\mathrm{D}_f(\overline{x}_{i-1}, \tilde{x}_{i-1})\}}. \tag{50}$$

Hence, we obtain the following:

$2\mathcal{D}^2$

$$\overset{(a)}{\geq} 2\sum_{i=1}^{k}\left(\eta_i^2\|\nabla f(\tilde{x}_i)\|^2 + \eta_i\,\mathrm{D}_f(\overline{x}_i; \tilde{x}_i)\right)$$

$$= \sum_{i=1}^{k}\left(\eta_i^2\|\nabla f(\tilde{x}_i)\|^2 + \eta_i\,\mathrm{D}_f(\overline{x}_i; \tilde{x}_i)\right)$$

$$+ \sum_{i=2}^{k+1}\left(\eta_{i-1}^2\|\nabla f(\tilde{x}_{i-1})\|^2 + \eta_{i-1}\,\mathrm{D}_f(\overline{x}_{i-1}; \tilde{x}_{i-1})\right)$$

$$\overset{(b)}{\geq} \sum_{i=1}^{k}\left(\eta_i^2\|\nabla f(\tilde{x}_i)\|^2 + \eta_i\,\mathrm{D}_f(\overline{x}_i; \tilde{x}_i)\right)$$

$$+ \sum_{i=2}^{k+1}\left(\frac{1}{(1+\gamma)^2}\eta_i^2\|\nabla f(\tilde{x}_{i-1})\|^2 + \frac{1}{(1+\gamma)}\eta_i\,\mathrm{D}_f(\overline{x}_{i-1}; \tilde{x}_{i-1})\right)$$

$$\overset{(c)}{\geq} \sum_{i=2}^{k}\left(\frac{\eta_i^2}{(1+\gamma)^2}\left(\|\nabla f(\tilde{x}_i)\|^2 + \|\nabla f(\tilde{x}_{i-1})\|^2\right) + \frac{\eta_i}{(1+\gamma)^2}(\mathrm{D}_f(\overline{x}_i; \tilde{x}_i) + \mathrm{D}_f(\overline{x}_{i-1}; \tilde{x}_{i-1}))\right)$$

$$\overset{(d)}{\geq} \sum_{i\in\mathcal{T}_2'(k)}\left(\frac{\eta_i^2}{(1+\gamma)^2}\left(\|\nabla f(\tilde{x}_i)\|^2 + \|\nabla f(\tilde{x}_{i-1})\|^2\right) + \frac{\eta_i}{(1+\gamma)^2}(\mathrm{D}_f(\overline{x}_i; \tilde{x}_i) + \mathrm{D}_f(\overline{x}_{i-1}; \tilde{x}_{i-1}))\right)$$

$$\overset{(e)}{\geq} \sum_{i\in\mathcal{T}_2'(k)}\left(\frac{\nu^2\lambda_i^2}{(1+\gamma)^2}\left(\|\nabla f(\tilde{x}_i)\|^2 + \|\nabla f(\tilde{x}_{i-1})\|^2\right) + \frac{\nu\lambda_i}{(1+\gamma)^2}(\mathrm{D}_f(\overline{x}_i; \tilde{x}_i) + \mathrm{D}_f(\overline{x}_{i-1}; \tilde{x}_{i-1}))\right),$$

where (a) uses Lemma 5; (b) uses line 11; (c) uses the fact that $\gamma > 0$; (d) uses the definition $\mathcal{T}_2'(k) = \mathcal{T}_2(k) \setminus \{1\}$; (e) uses the definition of $\mathcal{T}_2(k)$ in eq. (36) and the fact that $H_{i-2} \geq \eta_{i-2}$. Furthermore, using eq. (50), we can show that the following inequality holds for all $i \in \mathcal{T}_2'(k)$:

$$\max\left\{\lambda_i^2\|\nabla f(\tilde{x}_i)\|^2, \lambda_i^2\|\nabla f(\tilde{x}_{i-1})\|^2, \lambda_i\,\mathrm{D}_f(\overline{x}_i; \tilde{x}_i), \lambda_i\,\mathrm{D}_f(\overline{x}_{i-1}; \tilde{x}_{i-1}),\right\} \geq \frac{4}{81L_1^2}. \tag{51}$$

Hence, we obtain the following:

$$2\mathcal{D}^2 \geq \sum_{i\in\mathcal{T}_2'(k)}\left(\frac{\min\{\nu, \nu^2\}}{(1+\gamma)^2} \cdot \frac{4}{81L_1^2}\right) \overset{(a)}{\geq} \frac{4\min\{\nu, \nu^2\}}{81(1+\gamma)^2 L_1^2} \cdot |\mathcal{T}_2'(k)|,$$

where (a) uses the fact that $\gamma > 0$. It remains to use the inequality $|\mathcal{T}_2(k)| \leq 1 + |\mathcal{T}_2'(k)|$. $\quad\square$

## C.5 PROOF OF THEOREM 3

We prove the statement of Theorem 3 by induction over $K$. The base case, $K = 1$, is obvious. Now, we assume that eq. (40) holds for all $k \in \{0, \dots, K\}$, where $K \geq 1$, and prove the inequality in eq. (40) for $k = K + 1$. We consider the following cases:

**Case 1.** $K + 1 \in \mathcal{T}_2(K + 1)$ or $K + 1 \in \mathcal{T}_4(K + 1)$.

**Case 2.** $K + 1 \in \mathcal{T}_1(K + 1)$.

**Case 3.** $K + 1 \in \mathcal{T}_3(K + 1)$.

    **Case 3a.** $l(K + 1) = 0$.

    **Case 3b.** $l(K + 1) \in \mathcal{T}_1(K + 1)$.

    **Case 3c.** $l(K + 1) \in \mathcal{T}_2(K + 1)$.

**Case 1.** In this case, we can lower-bound $\sqrt{H_{K+1}}$ as follows:

$$\sqrt{H_{K+1}} \overset{(a)}{\geq} \sqrt{H_K}$$

$$\overset{(b)}{\geq} \frac{c}{\sqrt{L_0}} \cdot (K - |\mathcal{T}_2(K)| - |\mathcal{T}_4(K)| - 1)$$

$$\overset{(c)}{\geq} \frac{c}{\sqrt{L_0}} \cdot (K + 1 - |\mathcal{T}_2(K+1)| - |\mathcal{T}_4(K+1)| - 1),$$

where (a) uses Lemma 4; (b) uses the induction hypothesis in eq. (40); (c) uses the assumpttion $K + 1 \in \mathcal{T}_2(K+1) \cup \mathcal{T}_4(K+1)$ and the definition of $\mathcal{T}_2(K+1)$ and $\mathcal{T}_4(K+1)$ in eq. (36). This proves Case 1.

**Case 2.** In this case, we can lower-bound $\sqrt{H_{K+1}}$ as follows:

$$\sqrt{H_{K+1}} = \frac{1}{\sqrt{H_{K+1}} + \sqrt{H_{K-2}}}(H_{K+1} - H_{K-2}) + \sqrt{H_{K-2}}$$

$$\overset{(a)}{\geq} \frac{1}{2\sqrt{H_{K+1}}}(H_{K+1} - H_{K-2}) + \sqrt{H_{K-2}}$$

$$\overset{(b)}{\geq} \frac{1}{2\sqrt{H_{K+1}}}(\eta_{K+1} + \eta_{K-1}) + \sqrt{H_{K-2}}$$

$$\overset{(c)}{\geq} \sqrt{\frac{\eta_{K+1}\eta_{K-1}}{H_{K+1}}} + \sqrt{H_{K-2}}$$

$$\overset{(d)}{\geq} \sqrt{\frac{\nu \lambda_{K+1} H_{K-1}}{H_{K+1}}} + \sqrt{H_{K-2}}$$

$$\overset{(e)}{\geq} \sqrt{\frac{2\nu H_{K-1}}{9 L_0 H_{K+1}}} + \sqrt{H_{K-2}}$$

$$\overset{(f)}{\geq} \frac{\sqrt{2\nu}}{3(2+\gamma)\sqrt{L_0}} + \sqrt{H_{K-2}}$$

$$\overset{(g)}{\geq} \frac{3c}{\sqrt{L_0}} + \sqrt{H_{K-2}}$$

$$\overset{(h)}{\geq} \frac{3c}{\sqrt{L_0}} + \frac{c}{\sqrt{L_0}} \cdot (K - 2 - |\mathcal{T}_2(K-2)| - |\mathcal{T}_4(K-2)| - 1)$$

$$= \frac{c}{\sqrt{L_0}} \cdot (K + 1 - |\mathcal{T}_2(K-2)| - |\mathcal{T}_4(K-2)| - 1)$$

$$\overset{(i)}{\geq} \frac{c}{\sqrt{L_0}} \cdot (K + 1 - |\mathcal{T}_2(K+1)| - |\mathcal{T}_4(K+1)| - 1),$$

where (a) and (f) use Lemma 4; (b) uses line 11; (c) uses Young's inequality; (d) and (e) use the assumption $K + 1 \in \mathcal{T}_1(K+1)$ and the definition of $\mathcal{T}_1(K+1)$ in eq. (36); (g) uses the definition of $c$ in eq. (39); (h) uses the induction hypothesis in eq. (40); (i) uses the definition of $\mathcal{T}_2(k)$ and $\mathcal{T}_4(k)$ in eq. (36). This proves Case 2.

**Case 3a.** Since $l(K+1) = 0$, we have $\mathcal{T}_1(K+1) = \mathcal{T}_2(K+1) = \varnothing$. Hence, from eqs. (36) and (37) and line 11, we have $\eta_{K+1} = (1+\gamma)^{K+1}\eta_0$. Moreover, since $K + 1 \in \mathcal{T}_3(K+1)$ and $l(K+1) \notin \mathcal{T}_1(K+1)$, we have $K + 1 > m$ due to eqs. (36) and (37). Besides, since $\mathcal{T}_1(K+1) = \mathcal{T}_2(K+1) = \varnothing$, from eq. (36) we conclude that $\mathcal{T}_3(K+1) = \{m+1, \ldots, K+1\}$ and $\mathcal{T}_4(K+1) = \{1, \ldots, m\}$. Hence, we can lower-bound $\sqrt{H_{K+1}}$ as follows:

$$\sqrt{H_{K+1}} \overset{(a)}{\geq} \frac{\eta_0 \gamma (1+\gamma)^m}{4}(K - m + 2)^2$$

$$\overset{(b)}{\geq} \frac{c^2}{L_0}(K - m + 2)^2$$

$$\overset{(c)}{=} \frac{c^2}{L_0}(K + 2 - |\mathcal{T}_2(K+1)| - |\mathcal{T}_4(K+1)|)^2$$

$$\geq \frac{c^2}{L_0}(K + 1 - |\mathcal{T}_2(K+1)| - |\mathcal{T}_4(K+1)| - 1)^2,$$

where (a) is obtained similarly to eq. (42) in the proof of Theorem 2 in Appendix B.2; (b) uses the definition of $m$ in eq. (39); (c) uses the fact that $\mathcal{T}_4(K+1) = \{1, \ldots, m\}$ and $\mathcal{T}_2(K+1) = \varnothing$ as shown above. This proves Case 3a.

**Case 3b.** Using eqs. (36) and (37) and line 11, we can express $\eta_{K+1}$ as follows:

$$\eta_{K+1} = (1+\gamma)^{K+1-l(K+1)}\eta_{l(K+1)} = \frac{\nu H_{l(K+1)-2}\lambda_{l(K+1)}}{\eta_{l(K+1)-2}}. \tag{52}$$

Moreover, similar to eq. (44) in the proof of Theorem 2 in Appendix B.2, we can upper-bound $H_{K+1}$ as follows:

$$H_{K+1} \leq \frac{1}{\gamma}(1+\gamma)^{K-l(K+1)+3} H_{l(K+1)-1}. \tag{53}$$

In addition, similar to eq. (46) in the proof of Theorem 2 in Appendix B.2, and using the definition of $\mathcal{T}_1(k)$ in eq. (36), we can obtain the following inequality:

$$\left(\sqrt{H_{K+1}} - \sqrt{H_{l(K+1)-3}}\right)^2 \geq \frac{\nu}{18L_0\sqrt{\gamma(1+\gamma)(2+\gamma)^3}} \cdot \left((1+\gamma)^{(K-l(K+1))/2+1} - 1\right), \tag{54}$$

where $H_{-2} = H_0$, and similar to eq. (47) in the proof of Theorem 2 in Appendix B.2, we can obtain the following inequality:

$$\sqrt{H_{K+1}} - \sqrt{H_{l(K+1)-3}} \geq \frac{\sqrt{\nu}\gamma}{24\sqrt[4]{4\gamma(1+\gamma)^5(2+\gamma)^3 L_0^2}} \cdot (K - l(K+1) + 4). \tag{55}$$

Finally, after rearranging, we can lower-bound $\sqrt{H_{K+1}}$ as follows:

$$\sqrt{H_{K+1}} \overset{(a)}{\geq} \sqrt{H_{l(K+1)-3}} + \frac{\sqrt{\nu}\gamma}{24\sqrt[4]{4\gamma(1+\gamma)^5(2+\gamma)^3 L_0^2}} \cdot (K - l(K+1) + 4)$$

$$\overset{(b)}{\geq} \sqrt{H_{l(K+1)-3}} + \frac{c}{\sqrt{L_0}} \cdot (K - l(K+1) + 4)$$

$$\overset{(c)}{\geq} \frac{c}{\sqrt{L_0}} \cdot (l(K+1) - 3 - |\mathcal{T}_2(l(K+1)-3)| - |\mathcal{T}_4(l(K+1)-3)| - 1).$$

$$+ \frac{c}{\sqrt{L_0}} \cdot (K - l(K+1) + 4)$$

$$= \frac{c}{\sqrt{L_0}} \cdot (K + 1 - |\mathcal{T}_2(l(K+1)-3)| - |\mathcal{T}_4(l(K+1)-3)| - 1)$$

$$\overset{(d)}{\geq} \frac{c}{\sqrt{L_0}} \cdot (K + 1 - |\mathcal{T}_2(K+1)| - |\mathcal{T}_4(K+1)| - 1),$$

where (a) uses eq. (55); (b) uses the definition of $c$ in eq. (39); (c) uses the induction hypothesis in eq. (40); (d) uses the definition of $\mathcal{T}_2(k)$ and $\mathcal{T}_4(k)$ in eq. (36). This proves Case 3b.

**Case 3c.** Similar to eq. (46) in the proof of Theorem 2 in Appendix B.2, and using Lemma 6, we can obtain the following inequality:

$$\left(\sqrt{H_{K+1}} - \sqrt{H_{l(K+1)-3}}\right)^2 \geq \frac{\nu\lambda_{\min}}{4\sqrt{\gamma(1+\gamma)(2+\gamma)^3}} \cdot \left((1+\gamma)^{(K-l(K+1))/2+1} - 1\right)$$

$$= \frac{\nu\lambda_{\min}(1+\gamma)^{m/4}}{4\sqrt{\gamma(1+\gamma)(2+\gamma)^3}} \cdot \left((1+\gamma)^{(K-l(K+1)-m/2)/2+1} - \frac{1}{(1+\gamma)^{m/4}}\right)$$

$$\overset{(a)}{\geq} \frac{\nu}{18L_0\sqrt{\gamma(1+\gamma)(2+\gamma)^3}} \cdot \left((1+\gamma)^{(K-l(K+1)-m/2)/2+1} - 1\right)$$

$$\overset{(b)}{\geq} \frac{\nu}{18L_0\sqrt{\gamma(1+\gamma)(2+\gamma)^3}} \cdot \left((1+\gamma)^{(K-l(K+1))/4+1} - 1\right),$$

where $H_{-2} = H_0$, and (a) uses the definition of $m$ in eq. (39); (b) uses the fact that $K - l(K+1) \geq m$, which is implied by the assumptions $K + 1 \in \mathcal{T}_3(K+1)$ and $l(K+1) \in \mathcal{T}_2(K+1)$, and eq. (36). After taking the square root from both sides of the inequality, we obtain the following:

$$\sqrt{H_{K+1}} - \sqrt{H_{l(K+1)-3}} \geq \frac{\sqrt{\nu}}{3\sqrt[4]{4\gamma(1+\gamma)(2+\gamma)^3 L_0^2}} \cdot \sqrt{(1+\gamma)^{(K-l(K+1))/4+1} - 1}$$

$$\overset{(a)}{\geq} \frac{\sqrt{\nu}}{3\sqrt[4]{4\gamma(1+\gamma)(2+\gamma)^3 L_0^2}} \cdot \left((1+\gamma)^{(K-l(K+1)+4)/8} - 1\right)$$

$$= \frac{\sqrt{\nu}}{3\sqrt[4]{4\gamma(1+\gamma)^5(2+\gamma)^3 L_0^2}} \cdot \left((1+\gamma)^{(K-l(K+1)+4)/8+1} - (1+\gamma)\right)$$

$$\overset{(b)}{\geq} \frac{\sqrt{\nu}\gamma}{24\sqrt[4]{4\gamma(1+\gamma)^5(2+\gamma)^3 L_0^2}} \cdot (K - l(K+1) + 4),$$

where (a) uses the inequality $\sqrt{a} \geq \sqrt{a+b} - \sqrt{b}$; (b) uses Bernoulli's inequality. The rest of the proof is identical to the proof of Case 3b. □

## C.6 PROOF OF COROLLARY 3

We can upper-bound $\mathcal{D}^2$ as follows:

$$
\begin{aligned}
\mathcal{D} &\overset{(a)}{=} \sqrt{\max\left\{\tfrac{1}{\gamma\theta}, 2(1+\gamma), (1+2\theta)^2\right\}(\|x_0 - x^*\|^2 + (1+\gamma\theta)\eta_0^2\|\nabla f(x_0)\|^2)} \\
&\overset{(b)}{\leq} \sqrt{\max\left\{\tfrac{1}{\gamma\theta}, 2(1+\gamma), (1+2\theta)^2\right\}}\left(\|x_0 - x^*\| + \sqrt{(1+\gamma\theta)}\eta_0\|\nabla f(x_0)\|\right) \\
&\overset{(c)}{\leq} \sqrt{\max\left\{\tfrac{1}{\gamma\theta}, 2(1+\gamma), (1+2\theta)^2\right\}}\left(1 + (1+\gamma\theta)\eta_0 L_0 \cdot \tfrac{(\exp(L_1\|x_0 - x^*\|)-1)}{L_1\|x_0 - x^*\|}\right)\|x_0 - x^*\| \\
&\overset{(d)}{\leq} \sqrt{\max\left\{\tfrac{1}{\gamma\theta}, 2(1+\gamma), (1+2\theta)^2\right\}}(1 + (1+\gamma\theta)\eta_0 L_0 \exp(L_1\|x_0 - x^*\|))\|x_0 - x^*\| \\
&\overset{(e)}{\leq} \sqrt{\max\left\{\tfrac{1}{\gamma\theta}, 2(1+\gamma), (1+2\theta)^2\right\}}(2 + \gamma\theta)\|x_0 - x^*\| \\
&= \mathcal{O}(\|x_0 - x^*\|),
\end{aligned}
$$

where (a) uses eq. (33); (b) uses the inequality $\sqrt{a+b} \leq \sqrt{a} + \sqrt{b}$; (c) uses eq. (31); (d) uses the inequality $\exp(t) - 1 \leq t\exp(t)$, which is implied by the convexity of the function $t \mapsto \exp(t)$; (e) uses the assumption $\eta_0 L_0 \leq \exp(-L_1\|x_0 - x^*\|)$. It remains to combine Corollary 1, Lemma 8, and Theorem 3. $\qquad\square$

