# OpenReview forum: "Nesterov Finds GRAAL: Optimal and Adaptive Gradient Method for Convex Optimization"
_ICLR.cc/2026/Conference — ICLR 2026 Poster_

### Official Review · Reviewer_E3U6 · 2025-10-27

**Soundness:** 4
**Presentation:** 3
**Contribution:** 3
**Rating:** 8
**Confidence:** 4

**Summary:**

This paper proposes an accelerated adaptive method that can estimate the local Lipschitz constant of the gradients adaptively. It is based on a series of recent papers that use the difference of gradients or other empirical quantities to estimate the Lipschitzness of gradients. This work isn't the first to introduce an accelerated adaptive method like that, but it is the first one where the stepsize estimate can change by a constant non-decreasing factor. The authors prove convergence guarantees for $L$-smooth as well as $(L_0, L_1)$-smooth problems, claiming complexity improvements. The paper features strong theoretical results, but it doesn't present any numerical validation of the proposed method's performance.

**Strengths:**

The work is undeniably strong theoretically. The authors propose a method that is more adaptive than other accelerated methods, which is an important consideration for its practical performance. It is also clear that deriving the method and its convergence guarantees was no easy task. The algorithmic design can be insightful for future developments of other adaptive methods.

I also think the proposed method can be quite useful for deterministic convex optimization on its own. While convex optimization is a less popular area of applications these days, I still find it a valuable contribution since such problems arise in a lot of applications.

**Weaknesses:**

1. The method unfortunately requires the ability to compute functional values, unlike the prior work on GRAAL and AdGD.

2. In terms of complexity, there is no realy improvement upon the work of Li & Lan (2025). As the authors mention, the approach of Li & Lan (2025) is to use a line-search at the first iteration, which introduces only an additive term in the complexity, while guaranteeing that $\eta_0 L \ge 1$, so it doesn't affect the complexity in a multiplicative way. I don't see any limitation of this approach since both works require the use of functional values anyway. That being said, the proposed new method still may benefit a lot from updating the stepsize more adaptively in practice.

3. It's a bit unfortunate there is no numerical insights in this paper.

4. I found the discussion in Section 4.1 to be extremely hard to follow, especially Lemma 7. The four sets of indices are introduced without being properly explained, not to mention the third and the fourth sets use $l(k)$, whose meaning is even less obvious. It's completely unclear why these sets are introduced and what we should expect them to be. Moreover, it would be useful to remind the reader of the definition of $m$ around Lemma 8 as it's not used anywhere prior to that except for Theorem 2. It's also very confusing that the authors redefine $m$ in Theorem 3. To summarize, this section is extremely technical and would benefit from a rewrite that makes it more intuitive.

5. The way the proofs are structured in the appendix makes reading them a tedious and a strongly challenging endeavour. I wish the authors chopped it into pieces with intuitive steps, rather than just giving the bounds all as a single huge inequality. I just couldn't make myself go through all the steps to check them even though I wanted to understand the proof.

## Minor typos
"$f(x) \colon \mathbb{R}^d \to \mathbb{R}$" should be "$f \colon \mathbb{R}^d \to \mathbb{R}$"
"Hence, it is rarely used in practice." I don't think this statement is true, many convex problems are still solved with line-search based methods
"It is well known that AdaGrad has the complexity of GD in eq. (3) for L-smooth functions (Levy et al., 2018)". I think it needs a clarification here that the results of Levy are only for bounded domains. It is also worth mentioning that the update in (5) misses the $G^2$ constant inside the square root, which is important for non-smooth convergence results.
I think when introducing (6) and (7) it would be educational for the reader if the authors provided values of $\gamma$ and $\nu$ that have theoretical guarantees.
"Suh & Ma (2025) uses a stepsize" -> "Suh & Ma (2025) use a stepsize"
Some small issues in the citations: "Jonathan M Borwein" -> "Jonathan M. Borwein", "in banach spaces" -> "in Banach spaces", "barzilai and borwein gradient" -> "Barzilai and Borwein gradient", etc.

**Questions:**

1. The authors wrote that $\theta$, $\gamma$, and $\nu$ should satisfy equation (19), but it's not obvious from looking at the equation what possible values we can select from. Could you please provide at least explicit example in the paper?
2. The complexities in Table 1 for Algorithm 1 and for the method of Tyurin (2025) are different in the power of $L_1 \mathcal{D}$. Is the optimal complexity known for this term? Is the result of Vankv et al. (2024) optimal? Do the authors believe their complexity is worse because it's a limiation of the analysis or of the method itself?
3. When discussing AdGD for $(L\_0, L\_1)$ functions, the authors wrote "In fact, $\mathcal{D}$ also contains the initial objective function gap $f(x\_0) − f(x^\*)$, and hence, it may have an exponential dependency on the initial distance $\Vert x\_0 − x^\*\Vert$." However, it seems that the proof of Corollary 3 actually also shows an exponential dependency on $\Vert x_0 − x^*\Vert$, which is then removed by assuming $\eta\_0 L\_0 \le \exp(-L\_1 \Vert x\_0 - x^\* \Vert)$. Isn't it an unfair comparison?

---

> ### Author Response · Authors · 2025-11-21
>
> Thank you for the time and effort you put into reviewing the paper and for the high evaluation of our work. Below, we provide our answers to the questions and address the weaknesses.
>
> ## Weaknesses
>
> > The method unfortunately requires the ability to compute functional values, unlike the prior work on GRAAL and AdGD.
>
> It is true. However, it should not be much more restrictive, as a forward pass is typically still required before a backward pass when computing the gradient.
>
> > In terms of complexity, there is no realy improvement upon the work of Li & Lan (2025)...
>
> It is true that line search can help to avoid extra factors in the complexity according to Li & Lan (2025). However, this weakens the overall result of Li & Lan (2025) because the algorithm is no longer completely line search-free, even though it is only used in the first iteration. More importantly, this approach does not help in the $(L_0,L_1)$-smooth case. Even with the line search at the first iteration, the initial stepsize can be as small as $\eta_0 L_0\exp(L_1\Vert x_0 - x^*\Vert) \sim 1$, which implies the exponential factors in the complexity due to fundamental upper bounds $\eta_k \leq \mathcal{O}(\eta_0 k)$ and $H_k\leq \mathcal{O}(\eta_0 k^2)$.
>
> > It's a bit unfortunate there is no numerical insights in this paper.
>
> Although this is primarily a theoretical paper, adding preliminary experiments would indeed strengthen it. We plan to do this in a revised version of the paper.
>
> >I found the discussion in Section 4.1 to be extremely hard to follow, especially Lemma 7...
>
> Please refer to our separate message titled "Common Questions".
>
> > The way the proofs are structured in the appendix makes reading them a tedious and a strongly challenging endeavour...
>
> There are many ways to structure proofs, each with various advantages and disadvantages. Our method may indeed appear less intuitive. On the other hand, it allows us to make the proofs very compact (e.g., the proof of Theorem 1 is only three pages long) and maintain their rigor (every transition contains precise references). From our personal reviewing experience, we have encountered examples where the proofs contained significant volumes of text, attempting to create the illusion of clarity, and then skipped essential transitions or explanations, making them extremely hard to check and sometimes even incorrect. Overall, it's quite difficult to find a method that would incorporate all the advantages and avoid all the disadvantages. We also hope that our sketch of Section 4.1 (refer to our separate message titled "Common Questions") will help to improve the clarity of the proofs in Appendix C.
>
> > Minor typos
>
> Thank you for noticing the typos and inaccuracies. We will fix them in the revised version of the paper.
>
> ## Questions
>
> > The authors wrote that $\theta$, $\gamma$, and $\nu$ should satisfy equation (19)...
>
> Please refer to our separate message titled "Common Questions".
>
> > The complexities in Table 1 for Algorithm 1 and for the method of Tyurin (2025) are different in the power of...
>
> Unfortunately, we are not aware of any results on the optimality of this term. We believe that our analysis is more or less accurate: our analysis can even improve the additive term in the complexity result for AdGD by Gorbunov et al. (2024). We may include this in the revised version of the paper. However, we do not know if the gap between our Algorithm 1 and Tyurin (2025) can be closed. It may be unavoidable because Algorithm 1 is adaptive and the algorithm of Tyurin (2025) is not.
>
> > When discussing AdGD for $(L_0,L_1)$ functions, the authors wrote "In fact, $\mathcal{D}$ also contains the initial objective function gap $f(x_0) - f(x^*)$, and hence, it may have an exponential dependency on the initial distance." ...
>
> It is true that the results for Algorithm 1 and AdGD are very similar in this sense, as they both require a sufficiently small initial stepsize. We made our comment to highlight the incompleteness in the results by Gorbunov et al. (2024), rather than any fundamental flaw in AdGD.

---

> > ### Comment · Reviewer_E3U6 · 2025-11-24
> >
> > > However, it should not be much more restrictive, as a forward pass is typically still required before a backward pass when computing the gradient.
> >
> > This is only true in the specific case of deep learning or linear models, many other problems have derviatives that are computed independently of the functional values, e.g., optimizing a polynomial.
> >
> > > Li & Lan (2025)
> >
> > Thank you for explaning this, I think this discussion should be also included in the paper itself. The current writing that dismisses their work as having a bad complexity isn't fair.
> >
> > > Overall, it's quite difficult to find a method that would incorporate all the advantages and avoid all the disadvantages.
> >
> > I understand, all I want to communicate is that despite being interested in understanding the proof and key novelties, I was rather frustrated by how it was presented. Your approach is basically "all or nothing", one can either go through every single step or give up completely. AdGD paper, in contrast, tried to give some high-level intuition: the Lyapunov function and the reason it includes certain terms, the steps done to get gradient differences, it's all discussed in sufficient detail. Please note that I'm not claiming that your way of presenting the proof is wrong, all I want to say is that I personally prefer to get some high-level understanding even if I'm not reading every single step of the proof.

---

### Official Review · Reviewer_A1cJ · 2025-10-29

**Soundness:** 4
**Presentation:** 4
**Contribution:** 4
**Rating:** 10
**Confidence:** 5

**Summary:**

The paper allows exponential growth of the stepsize for adaptive accelerated gradient method. The results show true adaptivity of the method under the generalized smoothness.

**Strengths:**

The paper solves the issue with explicit (no backtracking) adaptive accelerated methods  -- they cannot increase the stepsize fast.

Detailed comparison with related works.

**Weaknesses:**

Despite experiments would be quite predictable it would be interesting to see a comparison with adproxgd and optimizers with backtracking beyond the standard smoothness. Not a real weakness, for the sake of comprehensiveness, the theoretical contribution is solid enough.

**Questions:**

-

---

> ### Author Response · Authors · 2025-11-21
>
> Thank you for the time and effort you put into reviewing the paper and for the high evaluation of our work.
>
> > Despite experiments would be quite predictable it would be interesting to see a comparison with adproxgd and optimizers with backtracking beyond the standard smoothness. Not a real weakness, for the sake of comprehensiveness, the theoretical contribution is solid enough.
>
> Thank you for the suggestion. We are planning to include preliminary experiments in the revised version of the paper.

---

### Official Review · Reviewer_BGKs · 2025-10-30

**Soundness:** 3
**Presentation:** 3
**Contribution:** 3
**Rating:** 6
**Confidence:** 3

**Summary:**

This paper develops an accelerated variant of the GRAAL algorithm that achieves near-optimal iteration complexity for convex optimization while maintaining adaptive stepsize capabilities. The main contribution is Algorithm 1 (Accelerated GRAAL), which incorporates Nesterov acceleration while adapting stepsizes to local curvature at a geometric rate. The authors conduct convergence analysis of the proposed algorithm for $L$-smooth and $(L_0, L_1)$-smooth functions. The key technical innovation is an "additional coupling step" that allows adaptive choice of acceleration parameters $\alpha_k$ without requiring predefined sequences or restricting stepsize growth.

**Strengths:**

1. The additional coupling step is a novel technique that elegantly enabling acceleration and adaptive step-size growth in one framework.

2. The analysis is layered and rigorous: a general potential-descent framework for convex $C^1$ objectives, followed by rate results for $L$-smooth and $(L_0,L_1)$-smooth settings. The bounds are near-optimal and adapt to unknown curvature, with only logarithmic dependence on the initial stepsize guess in the $L$-smooth case.

3. The paper clearly states its target question, the design obstacles, and how each algorithmic components (extrapolation, coupling, curvature estimation) addresses them.

**Weaknesses:**

1.  Lack of empirical validation. The paper presents no experiments. Small-scale benchmarks (logistic regression, least-squares, GLMs, robust convex losses) would verify geometric step-size growth in practice, overhead of computing $\Lambda$, and wall-clock speedups versus other algorithms.

2. The theory requires universal constants $(\theta, \gamma, \nu)>0$ satisfying equation (19), but the paper does not provide guidance on how to choose these parameters in practice. This is a significant practical limitation. The paper should provide at least one explicit set of valid parameters and discuss how parameter choices affect the hidden constants in the O(·) notation.

3. Results cover smooth convex minimization only; there is no composite/prox extension, constraint handling, or analysis with stochastic/inexact oracles.

**Questions:**

1. How does the algorithm perform empirically compared to AC-FGM, AdaNAG, and non-accelerated GRAAL?

2. Could you provide explicit $(\theta,\gamma,\nu)$ values that satisfy the required inequalities, and a recommended default $\eta_0$?

3. Because the method introduces several iterates ($x_k, \bar{x}_k, \hat{x}_k, \tilde{x}_k$) and requires multiple gradients for curvature estimation, the peak memory can exceed that of standard AGD. Have you considered memory-efficient implementation of the proposed algorithm?

---

> ### Author Response · Authors · 2025-11-21
>
> Thank you for the time and effort you put into reviewing the paper and for the high evaluation of our work. Below, we provide our answers to the questions and address the weaknesses.
>
> # Weaknesses
>
> > Lack of empirical validation. The paper presents no experiments. Small-scale benchmarks (logistic regression, least-squares, GLMs, robust convex losses) would verify geometric step-size growth in practice, overhead of computing $\Lambda$, and wall-clock speedups versus other algorithms.
>
> The experimental results would be quite predictable. In particular, Algorithm 1 would beat AC-FGM/Ada-NAG by a large margin on any problem with fast, e.g., exponential, growth of the local gradient Lipschitz constant (such as regression with an exponential loss function) due to the fundamental sublinear stepsize growth restriction in these algorithms. Additionally, Algorithm 1 would outperform AdGD even on $L$-smooth problems since algorithms with Nesterov acceleration typically outperform their non-accelerated counterparts significantly. In both cases, the computational overhead would not substantially alter the results. On the other hand, although we plan to add preliminary experiments in the revised version of the paper, it already contains strong state-of-the-art theoretical results and hence should not require any experiments, just like strong experimental papers do not require any theory.
>
> > The theory requires universal constants  satisfying equation (19), but the paper does not provide guidance on how to choose these parameters in practice. This is a significant practical limitation. The paper should provide at least one explicit set of valid parameters and discuss how parameter choices affect the hidden constants in the $\mathcal{O}(\cdot)$ notation.
>
> Please refer to our separate message titled "Common Questions".
>
> > Results cover smooth convex minimization only; there is no composite/prox extension, constraint handling, or analysis with stochastic/inexact oracles.
>
> Extending our results to the composite/proximal setting should be straightforward, since the baseline non-accelerated GRAAL can handle it. In the stochastic setting, developing such adaptive, even non-accelerated, methods is an interesting direction for future research that goes beyond the scope of this paper.
>
> # Questions
>
> > How does the algorithm perform empirically compared to AC-FGM, AdaNAG, and non-accelerated GRAAL?
>
> Please refer to our answer to weakness #1.
>
> > Could you provide explicit $(\theta,\gamma,\nu)$ values that satisfy the required inequalities, and a recommended default $\eta_0$?
>
> Please refer to our separate message titled "Common Questions". The recommended default $\eta_0$ is discussed, for instance, on lines 321-323.
>
> > Because the method introduces several iterates $(x_k,\overline{x}_k, \hat{x}_k, \tilde{x}_k)$ and requires multiple gradients for curvature estimation, the peak memory can exceed that of standard AGD. Have you considered memory-efficient implementation of the proposed algorithm?
>
> The maintenance of 4 sequences of iteration is not much more restrictive compared to baselines, such as the STM algorithm (Gasnikov & Nesterov, 2016), which uses 3 sequences of iterations. Computing $\lambda_k$ does not require computing multiple gradients, as it only uses the gradient $\nabla f(\tilde{x}_k)$ from the current and previous iterations. Of course, the requirement to compute additional function values and to store extra gradients/iterations will imply some overhead. However, this is the price for being the first algorithm of its kind: truly adaptive and provably accelerated. Simplifying the algorithm and reducing the memory and computational overhead is an interesting open question, which goes beyond the scope of this paper.

---

### Official Review · Reviewer_aKbR · 2025-11-04

**Soundness:** 3
**Presentation:** 3
**Contribution:** 3
**Rating:** 6
**Confidence:** 3

**Summary:**

This paper studies a GRAAL-based adaptive variant of Nesterov’s accelerated gradient method for smooth convex minimization. Unlike other recent adaptive methods, this approach allows the step size to grow linearly, which is particularly useful when the algorithm is initialized with a very small step size. This feature becomes even more important under the more general $(L_0,L_1)$-smoothness condition, where the local curvature may change at an exponential rate.

**Strengths:**

- As clearly summarized in Table 1, this paper proposes the first adaptive accelerated method that achieves the optimal rate under convexity and $(L_0,L_1)$-smoothness.

- Even under the standard smoothness assumption, the proposed adaptive accelerated method allows the step size to grow linearly, unlike existing methods, which is particularly beneficial when the initial step size is small.

**Weaknesses:**

- Although this is primarily a theoretical paper, including a simple toy experiment would make the contribution more informative. (This is not a requirement for acceptance, but a suggestion to strengthen the paper.)

- I would appreciate it if Section 4.1 provided a higher-level sketch of the main idea.

**Questions:**

- (10): The definition of $D_f$ is missing.
- L181: It seems that Option II can better exploit the properties of the objective function, but I recommend providing a more precise statement.
- L207: Could the authors clarify how the extrapolation enables adaptive capabilities?
- L226: The definition of $\Lambda$ is missing.
- L322: Wouldn't a small $\eta_0$ slow down the empirical performance?
- L461-462: The initial step size is required to satisfy $\eta_0 \le \frac{1}{L_0\exp(L_1||x_0-x^*||)}$, and the authors suggest choosing a very small value, which effectively makes the method free from hyperparameter tuning. The authors also note that a concurrent work by Tyurin (2025) provides a similar result but requires additional hyperparameter tuning. However, if one apply the same principle (choosing an extremely small or large constant), couldn't Tyurin's method also be regarded as tuning-free? This is what I inferred after reading Section 4 of Tyurin (2025). I would appreciate clarification on this point.

---

> ### Author Response · Authors · 2025-11-21
>
> Thank you for the time and effort you put into reviewing the paper and for the high evaluation of our work. Below, we provide our answers to the questions and address the weaknesses.
>
> ## Weaknesses
>
> > Although this is primarily a theoretical paper, including a simple toy experiment would make the contribution more informative...
>
> Thank you for the suggestion. We are planning to include preliminary experiments in the revised version of the paper.
>
> > I would appreciate it if Section 4.1 provided a higher-level sketch of the main idea.
>
> Please refer to our separate message titled "Common Questions".
>
> ## Questions
>
> > (10): The definition of $D_f$ is missing.
>
> We use the standard notation for the Bregman divergence $D_f(x,z) = f(x) - f(z) - \langle \nabla f(z), x-z\rangle$. We will include it in the revised version of the paper.
>
> > L181: It seems that Option II can better exploit the properties of the objective function, but I recommend providing a more precise statement.
>
> Unfortunately, it is hard to explain this at a high level without delving deeper into the analysis. More concretely, Option II contains the Bregman divergence, and to obtain certain lower bounds on the curvature estimator $\lambda_k$, one can exploit the negative Bregman divergence terms like the last one on line 715. This negative term comes from the analysis of the acceleration mechanism, and it would not be possible to obtain it for general variational inequality problems where there is no objective function.
>
> > L207: Could the authors clarify how the extrapolation enables adaptive capabilities?
>
> Unfortunately, once again, it is hard to explain this at a high level without considering the proof. In the case of variational inequalities, for which GRAAL was originally designed, it is anyway necessary to use some modifications (such as extrapolation or extragradient), because the standard gradient method may diverge even on simple bilinear min-max problems. For instance, we can consider a method for minimization problems with "gradient extrapolation":
> $$
> x_{k+1} = x_{k} - \eta_k g_k, \quad\text{where}\quad g_k = \nabla f(x_k) + \theta (\nabla f(x_k) - \nabla f(x_{k-1})),
> $$
> which is known as FoRB [a] aka OGDA [b]. From the convergence analysis, it turns out that if we try to make this algorithm adaptive using local curvature estimates with finite differences, computing the stepsize $\eta_k$ requires the estimate $L_{k+1} = \frac{\Vert \nabla f(x_{k+1}) - \nabla f(x_k) \Vert}{\Vert x_{k+1} - x_k \Vert}$. Hence, the algorithm cannot be implemented explicitly, as there is an obvious cyclic dependency between $\eta_k$ and $x_{k+1}$.
>
> GRAAL uses "iterates extrapolation" instead of "gradient extrapolation", i.e.,  $g_k = \nabla f(x_k + \theta(x_k - x_{k-1}))$. The key difference from FoRB/OGDA can be seen in the order in which the extrapolation and gradient computation are done: FoRB/OGDA computes gradients $\nabla f(x_k)$, $\nabla f(x_{k-1})$, and then performs the extrapolation $\nabla f(x_k) + \theta (\nabla f(x_k) - \nabla f(x_{k-1}))$, while GRAAL first performs the extrapolation of the iterates $\hat{x}\_k= x_k + \theta(x\_k - x_{k-1} ) $ and then computes the gradient at the extrapolated point $\hat{x}\_k$. From the analysis, it turns out that this "extrapolation before gradient computation" order allows the computation of the stepsize $\eta_k$ using the earlier estimate $L_k$ instead of $L_{k+1}$, which makes the method implementable. Unfortunately, once again, it is difficult to provide a more concrete explanation without delving further into the convergence proofs.
>
> [a] Malitsky & Tam (2020). A forward-backward splitting method for monotone inclusions without cocoercivity.
>
> [b] Daskalakis et al. (2017). Training GANs with optimism.
>
> > L226: The definition of $\Lambda$  is missing.
>
> $\Lambda(x;z)$ is defined in equation (11).
>
> > L322: Wouldn't a small $\eta_0$ slow down the empirical performance?
>
> With a small $\eta_0$, the stepsize is going to increase geometrically until it "catches up" with the current local curvature, both in theory and in practice. Hence, it will result in only a small logarithmic number of additional iterations at the beginning.
>
> > L461-462: The initial step size is required to satisfy...
>
> The key difference is that even with a very small $\eta_0$, our Algorithm 1 will automatically adapt the stepsize according to the local curvature at a fast geometric rate, without requiring prior knowledge of $L_0,L_1$. This will take only a small logarithmic number of iterations at the beginning. In contrast, Algorithm 2 of Tyurin (2025, Section 4) requires knowledge of $L_0,L_1$ at each iteration due to the stepsize rule on line 5 (refer to the definition of the function $\psi(x)$ on line 3 of Algorithm 2 and the definition of the function $\ell(x)$ in equation (2) by Tyurin (2025)). Hence, choosing excessively large values of $L_0, L_1$ will result in a very small stepsize and slow performance, both in theory and in practice.

---

### Author Response · Authors · 2025-11-21
**Common Questions**

In this message, we provide our comments on some common questions raised by reviewers.

## High-level sketch of Section 4.1

Below, we provide a high-level sketch of Section 4.1.
We hope that it will help to better understand the proof ideas from this section.
We will include a similar sketch in the revised version of the paper.

1. We split all iterations $1,\dots,K$ into four groups: $\mathcal{T}_1$, $\mathcal{T}_2$, $\mathcal{T}_3$, and $\mathcal{T}_4$, according to eq. (36):

    - $k \in \mathcal{T}_1 \cup \mathcal{T}_2$: the stepsize is computed using the local curvature estimate $\lambda_k$ (option 2 on line 11 of Algorithm 1).

    - $k \in \mathcal{T}_3 \cup \mathcal{T}_4$: geometric growth of the stepsize $\lambda_k$ (option 1 on line 11 of Algorithm 1).

2. $l(k)$ in eq. (37) denotes the latest iteration before $k$, where the stepsize is computed using the local curvature estimate $\lambda_k$. This implies that $l(k)+1,\dots,k$ is a segment of iterations with geometric growth of the stepsize, i.e., $\eta_k = \eta_{l(k)}\cdot (1+\gamma)^{k-l(k)}$. In addition $\eta_{l(k)} \geq \nu \lambda_{l(k)}\geq \nu \lambda_{\min}$ due to the stepsize rule on line 11 (option 2) and Lemma 6.

3. $k \in \mathcal{T}_1$ is considered "good": the stepsize is computed using a sufficiently large local curvature estimate $\lambda_k \gtrsim 1/L_0$. This case is similar to the proof for $L$-smooth functions.

4. $k \in \mathcal{T}_3$ is also considered "good":
    - Either $l(k) \in \mathcal{T}\_1$: the stepsize $\eta_{l(k)}$ is already sufficiently large and will grow further geometrically on the segment $l(k)+1,\dots, k$. This case is similar to the proof for $L$-smooth functions.
    - Alternatively, $l(k) \in \mathcal{T}\_2 \cup \{0\}$ and $k - l(k) > m$: the stepsize $\eta_{l(k)}$ may be as small as $\min\{\nu\lambda_{\min},\eta_0\}$, but the size of the segment of the stepsize geometric growth $l(k)+1,\dots, k$ is sufficiently large (greater than $m$) to compensate for this, i.e., $\eta_k \geq (1+\gamma)^m \min\{\eta_0, \nu\lambda_{\min}\}$, and $m$ defined in eq. (39) is sufficiently large.

5. Consequently, the iterations $k \in \mathcal{T}_2, \mathcal{T}_4$ are "bad", but their number is bounded by Lemma 8, using the inequalities from Lemma 5 and Lemma 7.

6. It remains to summarize the analysis of the "good" iterations and "ignore" the "bad" iterations in Theorem 3.


## Numerical constants $\theta$, $\gamma$, $\nu$

One can observe that the second inequality in eq. (19) holds, for instance, with $\theta = 4$ and $\gamma = \frac{11}{82}$. Hence, we can simply choose $\nu = \frac{\gamma}{4\theta(1+\gamma)^2}$ to ensure the first inequality in eq. (19). It is likely possible to improve these constants either by tightening the inequalities and transitions used in our proof or with the help of computer-aided analysis (e.g., Taylor, Adrien, and Francis Bach. "Stochastic first-order methods: non-asymptotic and computer-aided analyses via potential functions." Conference on Learning Theory. PMLR, 2019). For instance, Alacaoglu et al. (2023) wrote an entire paper, a large part of which is dedicated to improving the constants in the already existing GRAAL method (Malitsky, 2020).

We also wish to emphasize that we never aimed to "optimize" our convergence analysis to obtain the best possible values of $\theta$, $\gamma$, $\nu$. Rather, we focused on answering the question of whether it is possible, in principle, to develop an accelerated method that can adapt the stepsize at a sufficiently fast, geometric rate, just like its non-accelerated counterparts.

---

### Meta-Review · Area_Chair_zj19 · 2026-01-06

**Summary:**

The reviewers suggested that the contributions of this paper are good. They appreciate the new accelerated adaptive method with the ability to adapt to the local curvature of the objective function. The theoretical work is strong.

**Reviewer Concerns:**

Some reviewers suggested that small experiments would make the contributions more solid. The authors replied that they would include some preliminary experiments in the revision. Please add them for the publication.

One reviewer suggested that the author include the discussion about Li & Lan (2025) to the paper. In addition, some reviewers asked about the proof's intuition, and the authors added a sketch of the proof to clarify. Please include the proof sketch in the revision for publication.

**Reviewer Scores:**

One reviewer replied in the discussion. The initial scores are already good, so I think the final scores would remain the same.

---

### Decision · Program_Chairs · 2026-01-26

Accept (Poster)